# On the Measurement and Efficient Mitigation of Length Generalization Gaps in Large Language Models

## Abstract

Large Language Models (LLMs) typically train on short text due to the quadratic complexity of their self-attention architectures. As a result, their performance suffers drastically on inputs longer than those encountered during training, substantially limiting their applications in real-world tasks involving long contexts. In this paper, we rigorously establish an upper bound on length generalization in the measurement space and identify two length-related factors that limit performance. Our theory explains two recent observations: *(i)* out-of-distribution positions in longer contexts reduce length generalization, and *(ii)* fine-tuning on entire sequences is not necessary. Motivated by these insights, we propose *Virtual-context Learning (VCL)*, a flexible method that requires minimal modifications to most fine-tuning approaches. Experiments on various tasks show that *VCL* allows LLMs to generalize to $4\times$ context windows while retaining perplexity and improving performance on downstream tasks such as Passkey Retrieval and LongBench. *VCL* brings substantial efficiency improvements, reducing decoding time and memory usage by up to 50% compared with fine-tuning baselines.

## 1 Introduction

Large language models (LLMs) have recently advanced the state-of-the-art across various natural language processing tasks (Achiam et al., 2023; Team, 2025; Meta, 2024). They typically remain pre-trained on finite context windows primarily due to the computational overhead quadratic in the input lengths of their self-attention architectures Touvron et al. (2023a;b). As a result, their performance degrades significantly when applied to longer sequences (Ye et al., 2025; Bai et al., 2024; Chen et al., 2024; Kuratov et al., 2024), limiting their applicability in tasks that require long-range contexts, such as document retrieval, code generation, and story generalization (Bai et al., 2023; Zhang et al., 2024; Meta, 2024; Qin et al., 2024).

In this paper, we first provide a rigorous analysis of the upper bound for length generalization in pure self-attention using measure theory. Notably, attention matrices are discrete and of varying sizes across sequence lengths, preventing direct comparison or standard matrix operations. To address this question, we develop a measure-theoretic framework that maps token matrices and attention updates into a continuous probability space, which allows us to quantify distributional differences between short and long contexts and thereby analyze how attention distributions evolve as sequence length increases. Based on this framework, we identify two length-related factors that underlie generalization failures: *(i)* the shorter length $N$ (usually the pre-training context window) at a rate of $\sqrt{\ln N}$; *(ii)* the challenge of embedding distribution shifts for unseen or longer lengths. The increase in the test length can affect the distribution shift, increasing the distance between token embeddings and leading to failures in length generalization (Ye et al., 2025; Chen et al., 2023a).

Moreover, by connecting the bound factors of attention distribution distances to practical phenomena, we can systematically elucidate the mechanisms behind recent empirical findings (Clark et al., 2018; Chen et al., 2023a). *(i)* Longer contexts can diminish length generalization ability by introducing Out-of-Distribution (OOD) positions, which enlarges the distribution shift term in the bound. Existing position interpolation (PI) methods, such as Yarn (Peng et al., 2023), mitigate this effect by re-scaling longer position indices and adjusting the frequency basis to align with the scale inherited from the

pre-trained model during fine-tuning. *(ii)* Fine-tuning on entire long target sequences is not necessary, as only the short-length term appears in the bound directly. Instead, selecting a subset of important tokens from long sequences for fine-tuning yields comparable performance to using all tokens Fang et al. (2024); Hu et al. (2024). These selected tokens effectively mimic the long distribution and reduce the generalization gap, verifying the theoretical feasibility of extending context windows to even infinite lengths under limited computational resources.

Building on our theoretical insights, we introduce *Virtual-context Learning* (*VCL*), a simple and effective method that updates parameters only on long-length tokens. It trains solely on unseen or longer indices, reducing the quadratic cost of long inputs with minimal code changes in standard fine-tuning methods (Peng et al., 2023; Chen et al., 2023a). *VCL* can also optionally combine with PI to further alleviate token-distance effects caused by PE distribution shifts. Our experiments thoroughly evaluate *VCL* on a variety of tasks. On Proof-pile (Azerbayev et al., 2022) and GovReport (Huang et al., 2021), *VCL* facilitates length generalization against a wide range of PI series and PE-augmented methods up to $4\times$ length extension, retaining the language modeling perplexity and generation quality. *VCL* improves scores on downstream tasks including Passkey Retrieval (Mohtashami & Jaggi, 2023) and LongBench (Bai et al., 2023) which are two established benchmarks for long-context evaluation. We observe that *VCL* remains nearly 100% accuracy on OOD length on Passkey Retrieval and an average of 3.5 gain on LongBench. *VCL* also brings substantial efficiency improvements, reducing 50% training time and memory usage over full-length fine-tuning.

**In summary, our key contributions are:**

- From the theoretical perspective, we establish the upper bound of the length generalization performance under measurement space and pinpoint two length-related factors underlying the generalization failures: the shorter length term and the distribution shift distance of the different length sequences, opening the way to provably predict length generalization capabilities.

- Built upon our theoretical conclusion, we clarify the mechanisms behind recent empirical findings. Our results reveal that OOD positions in longer contexts reduce length generalization ability by increasing the distribution shift distance. Additionally, fine-tuning on entire sequences is not essential, as generalization performance is constrained solely by the short length, rather than directly by the long length.

- Based on our close observation, we propose a simple and effective method *Virtual-context Learning* (*VCL*) with minimal code changes during the standard fine-tuning phase, reducing the GPU usage and time consumption. Experiments across diverse benchmarks, including language modeling, passkey retrieval, and LongBench, demonstrate our *VCL* effectiveness in addressing length generalization challenges and further validating our theoretical insights.

## 2 RELATED WORK

**Length generalization** Length generalization remains a critical challenge in neural networks, as evidenced by extensive research (Delétang et al., 2022; Graves et al., 2016; Hupkes et al., 2020; Zhang et al., 2022). Even Transformer-based large language models (LLMs) (Chowdhery et al., 2023; Team, 2025), despite their sophisticated reasoning capabilities, struggle to process sequences that exceed their training length (Anil et al., 2022; Ye et al., 2025). Current approaches to improve length generalization, primarily focus on two aspects: enhancing positional encoding via fine-tuning methods (Chen et al., 2023b; Peng et al., 2023; Chen et al., 2023a) and optimizing input formats by zero-shot prompting (Quesnelle et al., 2023; Han et al., 2023b). Research has explored various positional encoding alternatives, including relative positional encodings that encode token-to-token distances (Dai et al.; McLeish et al., 2024), skip or randomized position encodings (Ruoss et al., 2023; Zhu et al., 2023), and weighted attention mechanisms as substitutes for position embeddings (Chi et al., 2022; Li et al., 2023; Press et al., 2021). Consequently, researchers have developed alternative data formatting techniques specifically for pretrained LLMs, including scratchpad methods and chain-of-thought approaches through in-context learning or fine-tuning (Xiao et al., 2023; Liu et al., 2024). However, these methods are challenging to implement during post-training and not robust (Zhou et al., 2024; Han et al., 2023a).

**Length generalization theory in transformer** Previous research (Bhattamishra et al., 2024; Liu et al., 2022; Vuckovic et al., 2020; Huang et al., 2024) has focused on the mathematical properties

of attention mechanisms, particularly examining properties such as Lipschitz continuity (Wu et al., 2024; Castin et al., 2023) or investigating the PE influence by modeling self-attention as a system of self-interacting particles (Vuckovic et al., 2020; Wu et al., 2024). Zhou et al. (2023) first posits the length generalization theory that transformers can achieve exact length generalization on algorithmic tasks solvable by simple programs RASP-L. Huang et al. (2024) further generalized it and provides a positive result for a specific idealized learning strategy under learnable PE. However, these studies (Han et al., 2023a; Press et al., 2022) often impose constraints on specific PEs and rely on numerous assumptions about input-output relationships. In this paper, we avoid making specific assumptions about PEs or input distributions; first investigate the influence of self-attention architectures on length generalization via the probability transformation.

## 3 PROBLEM SETUP

**(Measurement) Notation** We use the shorthand $[n] := \{1, 2, ..., n\}$. Let $\| \bullet \|_1$ be the 1-norm and $\mathbb{1}$ be the indicator function. For a real-valued function $f$, we use $\|f\|_{Lip} := \sup_{x \neq y} |f(x) - f(y)|/d(x, y)$ as the Lipschitz semi-norm. Throughout the analysis in the paper, we formalize the attention to be the kernel $\mathcal{A}$ in the probability space. Formally, let $(E, \mathcal{E})$ denote a subset of $\mathbb{R}^d$ endowed with its Borel $\sigma$-algebra, and $\mathcal{P}(E)$ be the space of all probability measures on $E$. We define the Markov kernels $\{\Psi_G(\nu)\}_{\nu \in \mathcal{P}(E)}$ by

$$\Psi_G(\nu)(\mathrm{d}x) := \frac{G(x)\,\nu(\mathrm{d}x)}{\nu(G)}, \nu \in \mathcal{P}(E),$$

where $G$ is a measurable function and $\nu(G) := \int_E G(x)\,\nu(\mathrm{d}x)$. For $A \in \mathcal{E}$, the kernel satisfies

$$\Psi_G(\nu)(x, A) := \int_A \Psi_{G(x, \cdot)}(\nu)(\mathrm{d}y).$$

For a function $G$ of two variables, $G : E \times E \to \mathbb{R}$, we let $\|G\|_{Lip, \infty} := \sup_{y \in E} \|G(\bullet, y)\|_{Lip}$ and $\|G\|_{\infty, Lip} := \sup_{x \in E} \|G(x, \bullet)\|_{Lip}$. In the attention mechanism, the measurable function $G$ is expressed in exponential form $G(x, y) = \exp(xy^T)$. A lookup kernel $L(x, dy)$ is a Markov kernel that maps $\mathcal{P}(X)$ to distributions on $\mathcal{P}(Y)$. Formal definitions are provided in Appendix E.

Furthermore, we will use the measurement-theoretic terminology (see Figure 1 for illustration):

**Definition 3.1 (Wasserstein Distance)** *Let $\mathcal{P}(E)$ be the set of probability measures, $\mu, \nu \in \mathcal{P}(E)$, and $\mathcal{C}(\mu, \nu)$ be the set of distributions on $(E \times E, \mathcal{E} \times \mathcal{E})$ with marginals $\mu, \nu$ on the first and second components, respectively. The Wasserstein distance between $\mu, \nu \in \mathcal{P}(E)$ is*

$$\mathbb{W}(\mu, \nu) := \inf_{\pi \in \mathcal{C}(\mu, \nu)} \iint_{E \times E} \|x - y\|_1 \pi(\mathrm{d}x, \ \mathrm{d}y). \tag{1}$$

**Attention Mechanism** Let $X = \{X_1, \ldots, X_N\} \in \mathbb{R}^{N \times d}$ be the token matrix, where $N$ is the length and $X_i \in \mathbb{R}^d$ represent the $i$-th token vector. The raw attention score matrix is computed as $Z = XW_Q(XW_K)^T/\sqrt{d_{QK}}$ where $W_Q, W_K \in \mathbb{R}^{d \times d'}$ are the query, key matrices respectively, and $d_{QK}$ is a temperature term to control the scale of raw attention scores. Without loss of generality, we assume $d_{QK} = 1$ in our analysis. To enforce attention, we create an attention matrix $A \in \mathbb{R}^{N \times N}$ where we normalize the attention scores among all tokens,

$$A_{ij} := \mathrm{softmax}(Z) = \frac{\exp(Z_{ij})}{\sum_{k=1}^N \exp(Z_{ik})}. \tag{2}$$

**Attention Update** For our analysis, we consider single-head self-attention networks (SAs). The layerwise update rule can be written as:

$$A^{(t)} = \mathrm{softmax}(X^{(t)} W_Q^{(t)} (X^{(t)} W_K^{(t)})^T / \sqrt{d_{QK}})$$
$$X^{(t+1)} := A^{(t)} X^{(t)} W_V^{(t)}, \tag{3}$$

where $W_V^{(t)} \in \mathbb{R}^{d \times d'}$ is the value matrix. For simplicity, throughout the paper, we assume that $d = d', W_Q = W_Q^{(t)}, W_K = W_K^{(t)}, W_V = W_V^{(t)}$. Yet the results can be easily generalized to the multi-layer case through iteration.

# 4 A MEASURE-THEORETIC FRAMEWORK FOR LENGTH GENERALIZATION

While the previous section formalizes the attention mechanism and its update rule, these formulations are insufficient for rigorously analyzing length generalization, since the attention outputs at different sequence lengths correspond to matrices of incompatible sizes that cannot be directly compared. To understand how attention distributions change as the sequence length grows, we need a framework that can measure distributional differences between short and long contexts; therefore, we map the attention mechanism into the continuous probability spaces (Vuckovic et al., 2020; Kim et al., 2021) by *(i)* representing queries, keys, and values as empirical measure mapping and *(ii)* the attention interaction as a Markov kernel transport. This abstraction allows us to use tools from measure theory, in particular the Wasserstein distance, to formally analyze how much attention distribution shifts under longer contexts and to derive theoretical bounds on length generalization.

## 4.1 ATTENTION KERNEL: A MEASURE THEORY VIEW

For simplicity, let $Q^{(t)} = X^{(t)}W_Q$, $K^{(t)} = X^{(t)}W_K$, and $V^{(t)} = X^{(t)}W_V$. Throughout this paper, we represent any $X \in \{Q^{(t)}, K^{(t)}, V^{(t)}\}$ by its empirical measure $m(X) = \{\delta_{X_i}\}_{i=1}^N$, regarded as a probability measure $\mathcal{P}(X)$. Here the Dirac measure $\delta_x(E)$ to a set $E$ assigns 1 if $x \in E$ and 0 otherwise. We then map the attention scores $A$ to the probability space via nonlinear Markov kernel transport on $m(Q^{(t)})$ and $m(K^{(t)})$ as follows:

**Proposition 4.1 (Attention Score Kernel)** *For the function $G(q, k) = exp(qk^T/d_{qk})$ where $d_{qk}$ is a temperature term to control the scale of raw attention scores, the attention score kernel $\mathcal{AS}$ is the family of Markov kernels $\{\Psi_G(\nu)\}_{\nu \in \mathcal{P}(E)}$. Specifically, for $m(Q^{(t)})$ and $m(K^{(t)})$, we have*

$$m(Q^{(t)})\Psi_G(m(K^{(t)})) = \frac{1}{N}\sum_{t=1}^N\sum_{s=1}^N \frac{G(Q_t^{(t)}, K_s^{(t)})}{\sum_{r=1}^N G(Q_t^{(t)}, K_r^{(t)})}\delta_{K_s^{(t)}}. \tag{4}$$

The detailed proof is provided in Appendix F.1. This formula converts the attention scores matrix $A$ for queries and keys into a conditional probability representation $\mathcal{AS}$ using Markov kernels, with the non-linear function $G$ expressed in exponential form. Finally, we present the attention as the interaction between attention scores and values in the form of a probability measure as follows:

**Proposition 4.2 (Self-attention Kernel)** *The self-attention kernel $\mathcal{A}_\mu$ is the composition of the attention score kernel $\mathcal{AS}$, the lookup kernel $L(k, dv) = \sum_{i=N} \mathbb{1}k = k_i\delta_{v_i}(dv)$ and the projection $\Pi$, defined for $x \in E$ and $\mu \in \mathcal{P}(E)$ as:*

$$\mathcal{A}_\mu(x, dz) := \Pi[\Psi_{G(x,\bullet)}(\mu)L](dz). \tag{5}$$

*Specifically, we have:*

$$m(X^{(t+1)}) \mapsto \left\{\delta_{Q_1^{(t)}}\mathcal{A}_{m(K^{(t)})}, \ldots, \delta_{Q_N^{(t)}}\mathcal{A}_{m(K^{(t)})}\right\} \tag{6}$$

*implements an attention mechanism.*

The detailed proof is provided in Appendix F.2.

Finally, we adopt the following assumptions in our analysis:

**Assumption 4.3** *There exists $C \in \mathbb{R}$ such that*

$$max_{t \in \mathbb{N}}\{\|G(Q^{(t)}, K^{(t)})\|_{Lip} + \|G(Q^{(t)}, K^{(t)})\|_\infty\} \leq C$$

**Assumption 4.4** *The empirical measures $\mathcal{P}(E) \in \{m(Q^{(t)}), m(K^{(t)}), m(V^{(t)})\}$ all have finite first moments that $\int_E \|x\|_1 d\mathcal{P}(x) < \infty$.*

**A** 4.3 states that the Lipschitz semi-norm and supremum norm of the exponential kernel $G$ remain uniformly bounded with respect to sequence length, which is essential for ensuring stable and efficient attention computation in practice (Kim et al., 2021; Castin et al., 2023), whereas **A** 4.4 guarantees that the Wasserstein distance between empirical measures is well-defined and finite for any $t > 0$, enabling the subsequent analysis of distributional shifts (Vuckovic et al., 2020).

## 4.2 LENGTH GENERALIZATION UPPER BOUND: 2 LENGTH-RELATED FACTORS

Building on the probabilistic representation of the attention kernel $\mathcal{A}$, we formalize length generalization as the Wasserstein distance between attention outputs under short and long contexts. As shown in Figure 1, if two token matrices convey similar semantics, their attention-induced measures should satisfy $\mathbb{W}(m(X_N^{(t+1)}), m(X_M^{(t+1)})) \approx 0$, reflecting the desired property of length generalization that representations remain close even when extending from short to long contexts ($M > N$). For example, the short sentence "The weather is hot." and the longer variant "In this hot weather, the sun shines brightly in the sky, the temperature

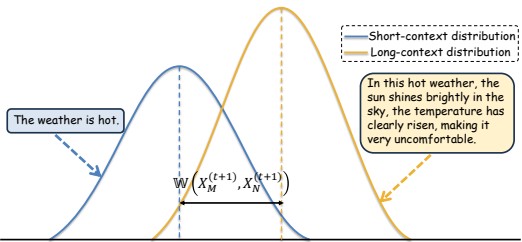

Figure 1: Intuition of our theoretical framework. Given two sentences with similar semantics, we measure the Wasserstein distance $\mathbb{W}$ between their attention-kernel outputs in probability space, which should yield similar representations.

has clearly risen, making it very uncomfortable." convey the same meaning and should yield similar distributions. This view provides a principled metric to quantify how model behavior shifts with length and underpins our theoretical bounds.

**Remark 4.5** *The Wasserstein distance $\mathbb{W}$ aligns with interpretable metrics for analyzing length generalization, such as JS distance, KL divergence, and perplexity. See Appendix C for empirical evidence under various settings in LLMs.*

Formally, based on the attention kernel $\mathcal{A}$, our goal is to measure $\mathbb{W}$ between the output of the attention kernel given two token matrices $X_N$ of length $N$ and input $X_M$ of arbitrary longer length $M > N$ with similar meanings. Finally, we prove the length generalization bound as follows:

**Theorem 4.6 (Length Generalization Upper Bound)** *Suppose two representations with different length $X_N^{(t)} = \left\{ X_1^{(t)}, \ldots, X_N^{(t)} \right\}$ and $X_M^{(t)} = \left\{ X_1^{(t)}, \ldots, X_M^{(t)} \right\}$, $N \leq M$, and the attention kernel $\mathcal{A}$. Let $\Pi$ be the usual projection. Then under **A** 4.3 and 4.4, for $\mu = m(X_N^{(t)})$ and $\nu = m(X_M^{(t)})$,*

$$\mathbb{W}\left( m(X_N^{(t+1)}), m(X_M^{(t+1)}) \right) \leq 2c(\Pi)c(L) \left[ \sqrt{d}\sqrt{\ln N + \frac{1}{2e}} \|G\|_{Lip} + \|G\|_{\infty} + \sqrt{d} + 2 \right] \mathbb{W}(\mu, \nu), \tag{7}$$

*where $c(\Pi), c(L)$ are some constants to be determined and the detailed expressions of $\|G\|_{\infty}$ and $\|G\|_{Lip}$ are seen in Appendix F.3.*

The detailed proof of Theorem 4.6 is provided in Appendix F.4. This result extends the findings of Vuckovic et al. (2020), which primarily analyzed the Lipschitz constant of attention but did not explicitly address how sequence length affects generalization. Under **A** 4.3, the terms $\|G\|_{\infty}$ and $\|G\|_{\text{Lip}}$ are intrinsic properties of the exponential similarity function $G$ defined on the space $E$ and remain bounded regardless of input length; similarly, $c(\Pi)$, $c(L)$, and the embedding dimension $d$ are independent of sequence size.

Consequently, **only two length-dependent factors govern generalization performance**:

*(i)* Shorter-length $\sqrt{\ln N}$: depends only on the shorter length $N$ and thus does *not* grow when evaluating on longer sequences, fortunately, which means that simply increasing test length does not worsen the bound through this factor.

*(ii)* Distribution shift distance $\mathbb{W}(\mu, \nu)$: captures how the probability distributions of short and long contexts diverge, which is the primary factor driving length generalization failures with the test length increase and the key target for improvement.

## 5   OUR THEORY-INSPIRED PROPOSAL: VIRTUAL-CONTEXT LEARNING

Having analyzed the Wasserstein distance of the attention kernel in probability space and established its dependence on sequence length, we now bridge theory with practice. Specifically, we examine *(i)* how positional encodings and their induced interpolation methods affect the distribution shift

term $\mathbb{W}(\mu, \nu)$, and **(ii)** whether full-length fine-tuning is an efficient way to extend context length. These insights motivate our proposed training strategy, *Virtual-context Learning*, which controls the effective training length while directly adapting to out-of-distribution positions.

## 5.1 A Close Look at Position Encodings and Full-length Fine-tuning

**Out-of-distribution (OOD) positions on longer contexts reduce length generalization.** A key source of length generalization failure lies in positional encodings: when extrapolated to unseen lengths, they distort the representation space and increase distributional mismatch (Kazemnejad et al., 2023; Gao et al., 2024). Our theoretical framework explains this phenomenon: OOD positions enlarge the distribution shift term $\mathbb{W}(\mu, \nu)$ in our bound, since the geometry of embeddings at unseen positions deviates from that observed during training. Notably, we follow Vuckovic et al. (2020); Castin et al. (2023) in encoding structural information (positions,

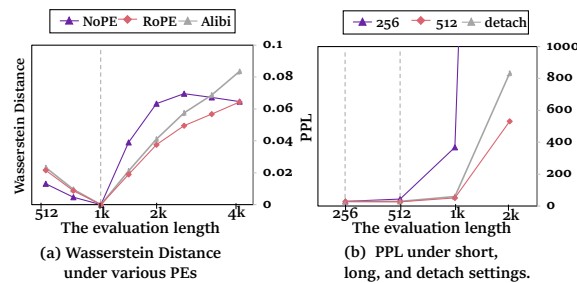

Figure 2: Implications of theorem. (a) OOD positions cause the distribution shift ($\mathbb{W}$ explosion). (b) Gradient backpropagation only on the longer length (detach) is comparable to full-length (512) parameter updates.

adjacency, semantics) in the token matrix $X$, which makes our theoretical framework applicable to diverse positional encodings, including RoPE (Su et al., 2022), Alibi (Press et al., 2022), and interpolation-based variants.

We empirically validate this explanation on CodeLlama with GovReport up to 4k tokens: Figure 2(a) shows that the average Wasserstein distance grows substantially once the evaluate sequence length exceeds the context window (1k). Existing position interpolation (PI) methods, such as Yarn (Peng et al., 2023), mitigate this effect by re-scaling longer position indices and adjusting the frequency basis to match the scale inherited from the pre-trained model during fine-tuning.

**Full-length fine-tuning on entire sequences is not necessary.** Recent works (Fang et al., 2024; Hu et al., 2024) show that selecting only a subset of important tokens from long sequences for fine-tuning yields comparable performance to using all tokens. Our theorem explains: perhaps counter-intuitively, increasing the training length $M$ does not inevitably harm length generalization, since only the short-length term appears in the bound. Thus, fine-tuning on carefully chosen tokens that reflect the long-context distribution is sufficient, rather than updating on the full sequence. In particular, OOD positions are the main source of discrepancy between short and long distributions; updating only on these positions can effectively mimic the long distribution and reduce the generalization gap.

To mitigate the effects of relative positional encodings and length extrapolation techniques applied in current base LLMs, such as Llama (Meta, 2024) and Qwen (qwe), we pre-train NanoGPT from scratch using the NoPE configuration. As empirical evidence shown in Figure 2(b), we establish a baseline with context windows of 256 and an oracle setting with 512, evaluating performance across different lengths using perplexity (PPL). Additionally, we present the results of pre-training on the RoPE versions in Appendix H.1. To verify that not all tokens are needed, we introduce a *detach* variant: in the 512-length setting, gradients from the first 215 tokens in each attention layer are frozen, and only the latter 215 tokens contribute to the loss. The results show that this detached method achieves performance on par with full-length (512) training, despite updating parameters only on the latter half of the sequence, consistent with our theoretical prediction.

## 5.2 Efficient Method: Not all tokens are needed for Training

Motivated by our previous analysis of length generalization, we propose *Virtual-context Learning* (*VCL*), which updates model parameters using only the latter, out-of-distribution tokens. Given a token matrix $X$, a language model $\theta$ is trained to maximize the conditional likelihood $P_\theta(X_i \mid$

Table 1: PPL across different methods. Blue marks the best. Calculation in loss means the token lengths used for loss during fine-tuning, while the evaluation context window size varies from 4k to 16k. *VCL* retains the low PPL even when the evaluation length $4\times$ than the calculation length.

| Methods | Calculation in Loss | GovReport | | | | | Proof-pile | | | | |
|---|---|---|---|---|---|---|---|---|---|---|---|
| | | 4k | 8k | 10k | 12k | 16k | 4k | 8k | 10k | 12k | 16k |
| Vanilla | - | 4.59 | >100 | >100 | >100 | >100 | 3.59 | 96.0 | >100 | >100 | >100 |
| RandPE | 4k | 5.12 | 8.0 | 9.63 | 11.06 | 14.0 | 4.19 | 5.28 | 5.91 | 6.47 | 8.0 |
| PoSE | 4k | 4.78 | 5.63 | 6.72 | 9.06 | 17.75 | 3.78 | 3.89 | 4.38 | 5.31 | 11.25 |
| Linear | 8k | 4.59 | 4.66 | 10.94 | 30.13 | >100 | 3.63 | 3.11 | 7.75 | 23.13 | 90.0 |
| NTK-aware | 8k | 4.59 | 4.66 | 7.34 | 20.13 | 74.5 | 3.59 | 3.16 | 4.97 | 14.69 | 60.0 |
| Yarn | 8k | 4.59 | 4.66 | 8.5 | 23.88 | >100 | 3.63 | 3.11 | 5.94 | 16.13 | 72.5 |
| without PI | | | | | | | | | | | |
| *VCL*-4k | 4k (8k-4k) | 6.22 | 5.81 | 6.06 | 6.88 | 10.25 | 5.41 | 4.28 | 4.16 | 4.38 | 5.72 |
| with Yarn (Peng et al., 2023) | | | | | | | | | | | |
| *VCL*-1k | 3k (4k-1k) | 4.63 | 4.66 | 4.59 | 4.56 | 4.59 | 3.66 | 3.16 | 3.03 | 2.92 | 2.72 |
| *VCL*-2k | 4k (6k-2k) | 4.94 | 4.84 | 4.94 | 4.81 | 4.97 | 3.92 | 3.36 | 3.17 | 3.08 | 2.92 |
| *VCL*-4k | 4k (8k-4k) | 4.66 | 4.66 | 4.59 | 4.56 | 4.63 | 3.72 | 3.16 | 3.03 | 2.92 | 2.73 |

$X_{<i}), i \in [N]$. Standard fine-tuning minimizes the loss over all tokens, while our objective becomes:

$$L_{VCL} = -\frac{1}{M-l} \sum_{i=l}^{M} \log P_\theta(X_i \mid X_{<i}), \tag{8}$$

where $l > 0$ specifies tokens ($0 \sim l$) used only in forward propagation without contributing to backpropagation. By excluding early tokens from gradient updates, *VCL* shortens the effective training length and reduces the distribution shift between short and long contexts, thereby improving length generalization. It can be seamlessly combined with position interpolation methods (e.g., Linear, NTK-aware, Yarn) to further minimize the embedding-space distance. Overall, *VCL* controls training length while adapting to unseen positions, lowering computation cost and mitigating overfitting to excessively long contexts.

# 6 EXPERIMENTS

**Experiment Settings.** We train the LLaMA-2-7B (Touvron et al., 2023b) with a context window of 4096, which does not expand the context window during the post-training phase like Qwen (qwe) or LLaMA-3 (Meta, 2024), to verify the effectiveness of our proposed methods without compromising generalization. We fine-tune all the baselines and our method on a dataset sourced from the Pile following (Zhu et al., 2023). The training uses a learning rate of $2 \times 10^{-5}$ with a linear scheduler, includes 10 warmup steps, and utilizes the AdamW optimizer with default hyperparameters, along with Flash Attention V2 (Dao, 2023). This process consists of 200 steps, employing a global batch size of 128 across 8 A100 GPUs with Deepspeed ZeRO stage 3 (Rajbhandari et al., 2020).

We choose the offset $l$ ranging from $\{1k, 2k, 4k\}$ and the target length $M$ from $\{4k, 8k, 10k, 12k, 16k\}$ where k is 1024. For evaluation, we use a single A100 GPU, making it possible to evaluate long documents of up to 16k tokens. We examine the ability of long text modeling on three tasks: language modeling with Perplexity (PPL), passkey retrieval (Mohtashami & Jaggi, 2023) with retrieval accuracy, and LongBench (Bai et al., 2023) with accuracy following with the length more than 8k.

**Baselines. Full-length**: We train the LLMs on the full target context length, serving as a baseline. **RandPE**: Ruoss et al. (2023) is initially designed to train an encoder-only model from scratch for length extrapolation. We include it for a comprehensive comparison. **PoSE**: Zhu et al. (2023) improve RandPE by dividing the original context window into two chunks and applying distinct skipping bias terms to manipulate the position indices of each chunk. **Linear & NTK-aware**: Chen et al. (2023b) involves a proportional down-scaling of the position index and NTK-aware methods defined in Peng et al. (2023) alter the base of RoPE, effectively modifying the rotational speed of each dimension of RoPE. **Yarn**: Peng et al. (2023) employs a ramp function to combine position interpolation methods such as Linear and NTK interpolation at varying proportions across different dimensions. We adopt the baseline implementation settings following Zhu et al. (2023).

Table 2: Accuracy on LongBench with extreme length (more than 8k). Blue marks best. We combined *VCL* with various position interpolation methods. Among the 6 categories of all the tasks, *VCL* performs best.

| Methods | Single-Doc QA | Multi-Doc QA | Summar- ization | Few-shot Learning | Synthetic Task | Code Completion | Avg |
|---|---|---|---|---|---|---|---|
| | | | LLama2-7b | | | | |
| Vanilla | 4.12 | 3.19 | 9.54 | 65.32 | 1.00 | 58.00 | 26.74 |
| FT(8k) | 9.33 | 7.62 | 15.11 | 62.47 | 3.08 | 56.94 | 28.58 |
| RandPE | 9.79 | 7.92 | 17.01 | 58.96 | 5.46 | 56.88 | 28.54 |
| PoSE | 11.85 | 8.38 | 16.91 | 62.53 | 4.07 | 53.01 | 28.93 |
| NTK | 11.39 | 7.47 | 15.79 | 61.53 | 4.01 | 56.75 | 28.88 |
| Yarn | 8.96 | 7.87 | 16.77 | 60.33 | 3.72 | 57.55 | 28.52 |
| | | | VCL | | | | |
| +Linear | 8.23 | 8.73 | 22.16 | 61.91 | 4.50 | 60.02 | 30.23 |
| +NTK | 10.73 | 8.77 | 17.63 | 61.83 | 4.96 | 59.20 | 29.85 |
| +Yarn | 13.20 | 8.15 | 16.87 | 62.94 | 6.12 | 55.73 | 29.92 |

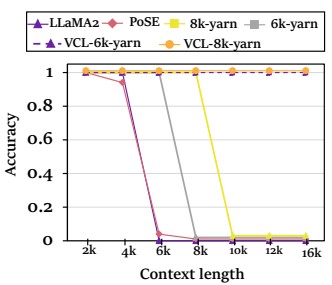

Figure 3: *VCL* shows best performance on the retrieval task, especially in a longer context than the fine-tuning.

## 6.1 LANGUAGE MODELING

First, we investigate the impacts of different fine-tuning methods on long sequence language modeling using the GovReport and Proof-pile datasets. Table 1 presents the PPL of scaling to evaluation length to 16k under vanilla, RandPE, PoSE, Yarn, and *VCL* with and without Yarn. In *VCL*, for the train length of 4k, we set the offset $l = 1k$; for lengths greater than 4k, we set $l = 2k, 4k$. We observe that there is little difference in perplexity between different methods for short lengths. However, for extended lengths, with approximately twice the fine-tuning length, we see a significantly larger gap in perplexities, indicating a notable difference in the ability of length generalization. We suppose it is because our *VCL* focuses on updating the parameters on the longer OOD position tokens, thereby mitigating the distribution shift between the long and short contexts, is consistent with our theoretical implications.

## 6.2 PASSKEY RETRIEVAL FOR EFFECTIVE CONTEXT WINDOW

To effectively measure the maximum distance that a token can attend to during the inference stage, we adopt the passkey retrieval test proposed by Mohtashami & Jaggi (2023). In this test, models are tasked with recovering a random passkey hidden within a lengthy document. The prompt template used for this task is presented in Figure 7(a) in the Appendix. We vary the prompt length from 2k to 16k. For each length, we conduct the passkey retrieval test 50 times, with a random passkey of 5 digits generated and placed at a random position inside the prompt. Figure 3 illustrates the results where 6k-yarn and 8k-yarn represent fine-tuning on 6k and 8k context windows with Yarn. VCL-6k-yarn and VCL-8k-yarn indicate fine-tuning on 6k and 8k context windows with VCL integrating Yarn. For the original, PoSE, and Yarn models, their retrieval accuracy rapidly drops to 0 with a maximum of 8k. In contrast, *VCL*-Yarn-6k / 8k models manage to maintain a high retrieval accuracy (nearly 100%) scaling to 16k. This indicates that models fine-tuned via *VCL* genuinely possess the capability to attend to all tokens both in the training length and the out-of-distribution length, verifying our theoretical insights.

## 6.3 EVALUATION ON LONG-CONTEXT BENCHMARK

To verify the complete performance of *VCL* in real-world scenarios, we further conduct an evaluation on the LongBench (Bai et al., 2023) with zero-shot setting. We fine-tune the llama2-7B on the GovReport dataset by employing *VCL* with constrained train lengths of 4k, combining the position interpolation tricks like Linear, NTK-aware, and Yarn. Table 2 shows that *VCL* generally outperforms other generalization methods, especially those with solely position interpolation methods, like Yarn. In the specific domain of QA, *VCL* remains competitive with or even surpasses the Full-length, demonstrating *VCL*'s superior length generalization ability. This substantiates the considerable potential of our theory insight for generalizing the context length to excel in long-context applications. Moreover, we adapt our *VCL* to other LLMs in the Appendix H.2.

## 6.4 ABLATION AND EFFICIENCY ANALYSIS

We now conduct detailed ablations to investigate the efficacy of the components in *VCL*:

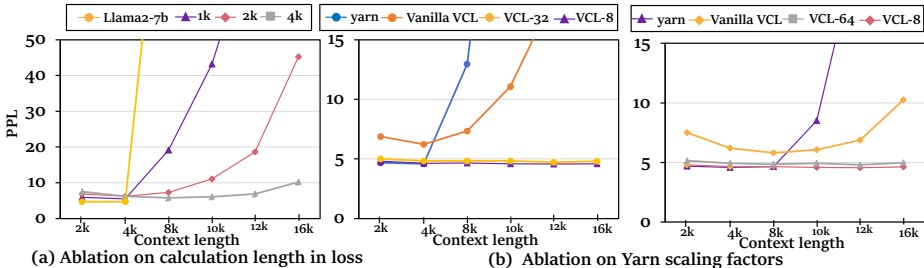

(a) Ablation on calculation length in loss      (b) Ablation on Yarn scaling factors

Figure 4: Ablation study of different hyperparameters of *VCL*. (a) varying the calculation length in loss $M - l$ from 1k to 4k, where $l$ is fixed with 4k. (b) **(left)** different yarn scaling factors under $M = 6k$. **(right)** different yarn scaling factors under $M = 8k$. yarn means fine-tuning on $M$ length and vanilla VCL means *VCL* without combining PI. *VCL-x* means *VCL* combined with Yarn with the scaling factors set to $x$.

**Enhance the calculation length in loss can be beneficial for the out-of-distribution length.** We add different lengths of context under our *VCL* to simulate the long context and test them on GovReport, ranging from 1k to 4k. We take the llama2-7b as the base model. In Figure 4(a), we discovered that a longer length improves the length generalization performance of our *VCL*, indicating the essentiality of the long context in *VCL*, aligning with common sense that fine-tuning on extended long contexts can improve the length generalization ability.

**Integrating the position interpolation with *VCL* can achieve greater length generalization ability.** To mitigate the Wasserstein Distance between the different length inputs induced by the positional encodings, we choose the classic PI strategy—Yarn to integrate with *VCL*. To enhance the generalization ability, we experiment on two train lengths 6k and 8k with offset $l = 2k, 4k$, respectively. The results in Figure 4(b) show that integrating *VCL* with the PI strategy outperforms using PI or *VCL* alone, with a significant decline in generalization ability when either is used separately. Furthermore, we observe that a larger extrapolation scale in Yarn does not improve length generalization ability, as noted in previous works (Chen et al., 2023a), a factor of 8 outperforms factors of 32/64. Therefore, the optimal factor must be carefully selected in practice.

***VCL* enhances the memory and computation efficiency .** We study the memory and time efficiency of *VCL* compared with Full-length fine-tuning (8k). The results of the experiment under the same training settings are shown in Figure 5, illustrating the memory and time consumption for 200 steps of Full-length versus *VCL*. Since *VCL* only updates the long-index tokens through the loss, it requires a small amount of memory and time for context extension, which is significantly lower compared to full-length fine-tuning. Consequently, we can confidently say that our proposed approach is both memory-efficient and time-efficient while enhancing the length generalization ability, further highlighting the superiority of our theory-based method *VCL*.

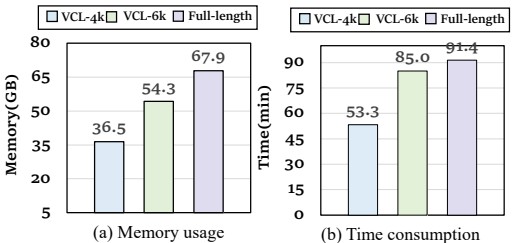

(a) Memory usage      (b) Time consumption

Figure 5: Efficiency of *VCL* compared to vanilla full-length fine-tuning. *VCL* effectively reduces the memory usage and time consumption for training.

## 7   CONCLUSION

In this work, we introduce a measure-theoretic framework to analyze length generalization in self-attention, revealing that the bound depends on the shorter length $N$ as $\sqrt{\ln N}$ and the geometric distance among input embeddings. Based on our theoretical framework, we provide the interpretation and empirical verifications for recent findings that out-of-distribution positions in longer contexts reduce length generalization, and that full-length fine-tuning on entire sequences is not necessary. Furthermore, we propose the theory-inspired *Virtual-context Learning* (*VCL*), a fine-tuning method that reduces computational costs and enhances length generalization by optimizing loss on long-token sequences. Experiments across diverse benchmarks validate *VCL* and support our theoretical findings, providing a principled understanding of enhancing generalization in LLMs.

**Reproducibility statement**    To ensure the reproducibility of our work, we have implemented several measures that are detailed throughout the paper and its supplementary materials. We commit to open-sourcing our code upon publication, which will allow others to replicate our experiments easily. In the appendix, we provide comprehensive explanations of the theoretical assumptions underpinning our results, along with complete proofs to substantiate our claims. Additionally, we have included a thorough description of the datasets used in our experiments, along with the specific data processing steps undertaken. We believe these resources will facilitate reproducibility and encourage further exploration of our findings.

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

# Appendix

Contents

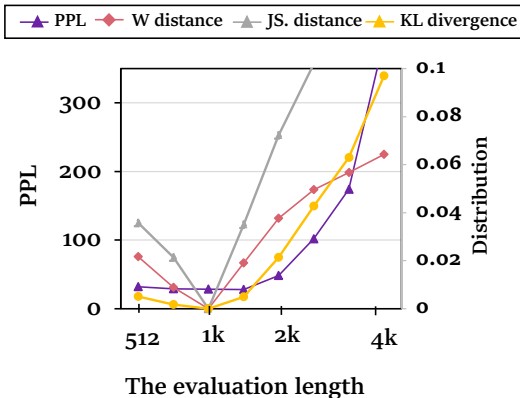

Figure 6: Comparasion of Wasserstein distance and other metrics for length generalization. They have the similiar test behaviors on evaluation length.

## A   LLM USAGE STATEMENT

**LLM Usage Statement**   We employed a large language model (LLM) as a supportive tool during the preparation of this manuscript. The LLM's involvement was strictly limited to enhancing the clarity and readability of the text, which included tasks such as grammar correction, spelling checks, rephrasing for conciseness, and refining sentence structure. The LLM did not contribute to any core research elements, including method ideation, theoretical derivation, experimental design, or result analysis. The authors have carefully reviewed all suggested edits and assume full responsibility for the content presented in this paper.

## B   LIMITATIONS AND BROADER IMPACTS

**Limitations.** Due to limited computational resources and time, the proposed method has not been evaluated on texts with even larger lengths, such as ranging from 100k 1M. *VCL* is designed for fine-tuning, it can be adapted to pre-training, where it may offer greater effectiveness. Preliminary experiments on NanoGPT in Appendix H.1 suggest its potential, but due to resource limitations, we do not provide results on larger-scale LLMs, leaving this for future work.

**Broader Impacts.** In this paper, we first propose a measure theory to quantify the length generalization bound of LLMs. This enables a deep understanding of the length generalization failure and thus points out the shortcomings of a wide range of length generalization methods. Based on our in-depth theoretical analysis, we reveal the mechanisms of empirical findings and propose a plug-and-play method *Virtual-context Learning*. We hope that our work can provide new insights and the underlying mechanisms of length generalization. For social impact, this work has a certain impact on the controllable and explainable AGI.

## C   WASSERSTEIN DISTANCE AND OTHER METHODS

A common approach to analyze length generalization in LLMs is to visualize attention or output distributions and quantify their shift using metrics such as Jensen–Shannon (JS) distance or KL divergence (Zhong et al., 2024; Cheng et al., 2025). We adopt the Wasserstein distance $\mathbb{W}(\mu, \nu)$ as our primary measure, which is consistent with these metrics and closely linked to the widely used perplexity (PPL) (Quesnelle et al., 2023; Ye et al., 2025; Bai et al., 2024), defined as the inverse geometric mean of token probabilities. We empirically support this by plotting JS distance, KL divergence, PPL, and Wasserstein distance against context length on CodeLlama (Figure 6), all showing similar upward trends: for unseen long contexts, the distances grow with length. Results under different metrics and PEs are provided in Appendix H.3, confirming that our measurement-based approach and theoretical framework generalize across settings.

# D MORE RELATED WORKS

**Position Interpolation methods**  While existing PI techniques can reduce the PE shift and thus alleviate the second challenge, fine-tuning on longer sequences is not necessary, as it theoretically extends the training length term of the upper bound. Extensive efforts have been devoted to addressing this length generalization challenge. Position Interpolation (PI) series methods such as Yarn (Peng et al., 2023) and CLEX (Chen et al., 2023a) have been proposed to extend the pretrain context windows, which fine-tune on the target length by position indices scaling and frequency basis scaling, hoping to avoid model failures due to unseen position embeddings (PEs). Meanwhile, position-augmented fine-tuning methods such as RandPE (Ruoss et al., 2023) and PoSE (Zhu et al., 2023) simulating longer inputs within a fixed window by adjusting position indices. Although position-augmented methods can reduce the memory overhead compared to PI methods, they disrupt local sentence structures and leave a significant gap in understanding token relationships across the sequence.

**Fine-tuning LLMs for longer context.**  Recently, a variety of length generalization methods have been developed to extend the context window of pre-trained LLMs (Fang et al., 2024; Chen et al., 2023a; Peng et al., 2023; Chen et al., 2023b). A straightforward approach is to fine-tune these models on target extensive texts. To mitigate the distribution shift of LLMs, Chen et al. (2023b) first down-scaled position indices to match original context size through Linear Position Interpolation. Subsequently, various Positional Interpolation (PI) strategies have been introduced, including NTK-aware (Peng & Quesnelle, 2023), Yarn (Peng et al., 2023), and CLEX (Chen et al., 2023a). Meanwhile, position-augmented fine-tuning methods such as RandomPE (Ruoss et al., 2023), FIRE (McLeish et al., 2024), and POSE (Zhu et al., 2023) simulate longer inputs within a fixed window by adjusting position indices. Although position-augmented methods can reduce the memory overhead compared to PI methods, they disrupt local sentence structures and leave a significant generalization gap in understanding token relationships across the sequence. Besides, LongLora (Chen et al., 2023c) proposes to shift short attention to approximate full attention. All of these methods seek to extend the context window length in more efficient fine-tuning ways.

**Self-attention.**  Formally, self-attention is defined as follows: $W_Q, W_K, W_V \in \mathbb{R}^{d \times d'}$ are the query, key, and value matrices respectively, and $\tau$ be the temperature, self-attention is:

$$\mathrm{SA}(X_i W_Q, X W_K, X W_V) := \sum_{j=1}^{N} \mathrm{softmax}(X_i W_Q, X_j W_K) X_j W_V, \tag{9}$$

where $\mathrm{softmax}(X_i W_Q, X_j W_K)$ is defined as

$$\mathrm{softmax}(X_i W_Q, X_j W_K) := \frac{\exp(X_i W_Q (X_j W_K)^T / \tau \sqrt{d})}{\sum_{s=1}^{N} \exp(X_i W_Q (X_s W_K)^T / \tau \sqrt{d})}. \tag{10}$$

For simplicity, throughout the paper, we assume that $d = d'$, $Q = X W_Q$, $K = X W_K$, and $V = X W_V$, thus $\mathrm{SA}(X_i W_Q, X W_K, X W_V)$ can be expressed as $\mathrm{SA}(Q_i, K, V)$. Moreover, while $\mathrm{SA}(Q_i, K, V)$ is defined point-wise for a given token $X_i$, it is almost always used to process a set of tokens $X = \{X_1, \ldots, X_N\}$ in parallel. Thus, we write the sequence-wise self-attention $\mathrm{SA}(Q, K, V) := \{\mathrm{SA}(Q_i, K, V))\}_{i=1}^{N}$.

# E MEASURE THEORY NOTATIONS

We will use the following constructions from measure theory: Let $(E, \mathcal{E})$ denote a subset of $\mathbb{R}^d$ endowed with its Borel $\sigma$-algebra, and $\mathcal{P}(E)$ be the space of all probability measures on $E$.

**Definition E.1 (Dirac Measure)** *Denote $\mathcal{P}_\delta(E) := \{\delta_x \mid x \in E\}$ be the subset of Dirac measures in $\mathcal{P}(E)$, where the Dirac measure $\delta_x$ is a measure defined by:*

$$\delta_x(E) = \begin{cases} 1, & \text{if } x \in E, \\ 0, & \text{if } x \notin E. \end{cases}$$

**Definition E.2 (Empirical Measure Mapping)** *For $X = \{x_1, \ldots, x_N\} \subset E$, the empirical measure associated with $X$ is the probability measure $m(X)$ defined by:*

$$X \mapsto m(X) := \frac{1}{|N|} \sum_{t=1}^{N} \delta_{x_t}. \tag{11}$$

**Definition E.3 (Markov Kernel)** *A Markov kernel is an $E$-indexed family of probability measures $M(x, \mathrm{d}y) \in \mathcal{P}(E)$ such that $\forall A \in \mathcal{E}, x \mapsto M(x, A)$ is measurable.*

**Definition E.4 (Lookup Kernel)** *Let the key space $(\mathcal{K}, \mathbb{K})$ and value space $(\mathcal{V}, \mathbb{V})$ be measurable, lookup kernel is a Markov kernel, $L : \mathcal{K} \times \mathbb{V} \to [0, 1]$, also denoted $L(k, \mathrm{d}v)$, that maps keys to distributions on values. When the mapping is a deterministic lookup table, we have $L(k, \mathrm{d}v) = \sum_{i=1}^{N} \mathbb{1}_{k=k_i} \delta_{v_i}(\mathrm{d}v).$*

**Definition E.5 (Moment Encoding and Subspace)** *Let $F : E \to E' \subset \mathbb{R}^l$ be a measurable map representing an $l$-dimensional feature map. (1) We say that a measure $\mu \in \mathcal{P}(E)$ encodes a moment vector $f \in E'$ with respect to function $F$ if $\mu(F) = f$. (2) Suppose we have identified an injective mapping $f \mapsto \nu_f \in \mathcal{P}(E)$ such that $\nu_f$ encodes the moment vector $f$ w.r.t. $F$. Then we denote by $\mathcal{F}_F = \{\nu_f \mid f \in E'\}$ the moment subspace of all such distributions.*

**Definition E.6 (Moment Projection)** *For simplify, we omit the subscript $F$ in $\mathcal{F}_F$. Let the moment projection $\Pi_{\mathcal{F}} : \mathcal{P}(E) \to \mathcal{F}$ be $\Pi_{\mathcal{F}}(\mu) = \nu_{\mu(F)}$ where $\Pi_{\mathcal{F}}(\mu)$ is the unique measure in $\mathcal{F}$ that encodes the moments $f := \mu(F)$*

**Definition E.7 (Finite First Moment)** *We say that a measure $\mathcal{P}(E)$ has a finite first moment if $\int_E \|x\|_1 \mathrm{d}\mathcal{P}(x) < \infty$ .*

Initially, the input matrices $Q, K, V$ are mapped to the measurable probability space $\mathcal{P}_\delta(\bullet)$ via empirical measure mapping. Through the softmax kernel, $\mathcal{P}_\delta(\bullet)$, a finite-dimensional feature space, is transformed into the infinite-dimensional probability space $\mathcal{P}(\bullet)$. To recover the matrix-level self-attention outputs, $\mathcal{P}(\bullet)$ must be projected back onto the finite-dimensional feature space $\mathcal{P}_\delta(\bullet)$, which is achieved using the moment projection. We claim that the averaging to input values is accomplished by the moment projection $\Pi := \Pi_{\mathcal{F}}$ described in Definition E.6, with $\mathcal{F} = \mathcal{P}_\delta(Q)$ and $F(x) = x$.

# F  PROOF

## F.1  PROOF OF PROPOSITION 4.1

Consider a "query" representation $\delta_q$ and "key" representations $K = \{\delta_{k_1}, \ldots, \delta_{k_N}\}$ and the empirical measure $m(K)$. The softmax kernel models the interaction between $q$ and $K$ using the left-action of the Markov kernels $\Psi_G(m(K))$ on the Dirac measure $\delta_q$ induced by integration:

$$\delta_q \Psi_G(m(K)) = \int \delta_q(\mathrm{d}q') \Psi_{G(q', \bullet)}(m(K)) = \Psi_{G(q, \bullet)}(m(K)) = \sum_{s=1}^{N} \frac{G(q, k_s)}{\sum_{r=1}^{N} G(q, k_r)} \delta_{k_s}.$$

Furthermore, given set of queries $Q = \{\delta_{q_1}, \ldots, \delta_{q_M}\}$, we can leverage the linearity of integration to model the interaction between the two sets of representations $Q$ and $K$ using the same principle:

$$m(Q) \Psi_G(m(K)) = \frac{1}{M} \sum_{t=1}^{M} \int \delta_{q_t}(\mathrm{d}q) \Psi_{G(q, \bullet)}(m(K))$$

$$= \frac{1}{M} \sum_{t=1}^{M} \Psi_{G(q_t, \bullet)}(m(K)) = \frac{1}{M} \sum_{t=1}^{M} \sum_{s=1}^{N} \frac{G(q_t, k_s)}{\sum_{r=1}^{N} G(q_t, k_r)} \delta_{k_s}.$$

## F.2 Proof of Proposition 4.2

Using the proposition 4.1, for $q \in \mathcal{Q}$, we have:

$$[\Psi_{G(q,\bullet)}(m(K))L](dv) = \sum_{j=1}^{N} \int \frac{G(q,k_j)}{\sum_{p=1}^{N} G(q,k_p)} \delta_{k_j}(dk)L(k,dv) = \sum_{j=1}^{N} \frac{G(q,k_j)}{\sum_{p=1}^{N} G(q,k_p)} \delta_{v_j}(dv).$$

Applying $\Pi$ thus yields: $A_{m(K)}(q,dv) = \delta_{\sum_{j=1}^{N} \frac{G(q,k_j)}{\sum_{p=1}^{N} G(q,k_p)} v_j}(dv)$. Using the (linear) left-action of this kernel on $\delta_{q_t}$, we then obtain:

$$\delta_{q_t} A_{m(K)}(dv) = \int \delta_{q_t}(dq) A_{m(K)}(q,dv) = \delta_{\sum_{j=1}^{N} \frac{G(q_t,k_j)}{\sum_{p=1}^{N} G(q_t,k_p)} v_j}(dv).$$

Plugging in the definition of $G$ and using the usual bijection $\delta_x \leftrightarrow x$ concludes the proof.

## F.3 Proof of Proposition F.1

**Proposition F.1** *Let $G(q,k) = exp(q^\top k/\tau)$, then we have*

$$\|G\|_{Lip} = \frac{1}{\tau} \sup_{q,k} \|q\| \|k\| \exp\left(\frac{\|q^\top k\|}{\tau}\right), \tag{12}$$

$$\|G\|_\infty = \sup_{q,k} \exp\left(\frac{\|q\| \|k\|}{\tau}\right). \tag{13}$$

To analyze the Lipschitz norm and supremum norm of the function $G(q,k) = \exp(q^\top k/\tau)$, we proceed as follows.

**Step 1: Compute the Lipschitz norm $\|G\|_{Lip}$**

By definition, the Lipschitz norm of $G$ is given by

$$\|G\|_{Lip} = \sup_{q,k \neq q',k'} \frac{|G(q,k) - G(q',k')|}{\|(q,k) - (q',k')\|}. \tag{14}$$

We first compute the gradient of $G(q,k)$:

$$\nabla G(q,k) = \frac{1}{\tau} G(q,k)(k,q). \tag{15}$$

Thus, the operator norm (i.e., the Lipschitz constant) is given by

$$\|\nabla G(q,k)\| = \sup_{q,k} \frac{1}{\tau} \|G(q,k)(k,q)\|. \tag{16}$$

Since $G(q,k) = \exp(q^\top k/\tau)$, we obtain

$$\|G\|_{Lip} = \frac{1}{\tau} \sup_{q,k} \|q\| \|k\| \exp\left(\frac{\|q^\top k\|}{\tau}\right). \tag{17}$$

**Step 2: Compute the supremum norm $\|G\|_\infty$**

By definition, the supremum norm is given by

$$\|G\|_\infty = \sup_{q,k} |G(q,k)|. \tag{18}$$

Since $G(q,k) = \exp(q^\top k/\tau)$, we take the supremum over all possible values of $q^\top k$, leading to

$$\|G\|_\infty = \sup_{q,k} \exp\left(\frac{\|q\| \|k\|}{\tau}\right). \tag{19}$$

This completes the proof.

## F.4 Proof of Theorem 4.6

To prove the theorem, we need the following lemma.

**Lemma F.2** *1. Suppose that $\Phi, \Gamma : \mathcal{P}(E) \to \mathcal{P}(E)$ are (possibly nonlinear) mappings. Then*

$$\tau(\Phi \circ \Gamma) \leq \tau(\Phi)\tau(\Gamma).$$

*2. Suppose $K : E \times \mathcal{E} \to [0, 1]$ is an integral kernel. Then*

$$\tau(K) = \sup_{x \neq y} \frac{\mathbb{W}_1(K(x, \bullet), K(y, \bullet))}{d(x, y)}.$$

*3. Suppose $K_1, K_2 : E \times \mathcal{E} \to [0, 1]$ are two integral kernels and $\nu \in \mathcal{P}(E)$. Then:*

$$\mathbb{W}_1(\nu K_1, \nu K_2) \leq \int \nu(dx) \mathbb{W}_1(K_1(x, \bullet), K_2(x, \bullet)).$$

**Proof.**

1. This is a standard result on Lipschitz constants. We include it for completeness:

$$
\begin{aligned}
\tau(\Phi \circ \Gamma) &= \sup_{\mu \neq \nu} \frac{\mathbb{W}_1(\Phi \circ \Gamma(\mu), \Phi \circ \Gamma(\nu))}{\mathbb{W}_1(\mu, \nu)} \\
&= \sup_{\mu \neq \nu} \frac{\mathbb{W}_1(\Phi \circ \Gamma(\mu), \Phi \circ \Gamma(\nu))}{\mathbb{W}_1(\Gamma(\mu), \Gamma(\nu))} \frac{\mathbb{W}_1(\Gamma(\mu), \Gamma(\nu))}{\mathbb{W}_1(\mu, \nu)} \\
&\leq \sup_{\eta \neq \gamma} \frac{\mathbb{W}_1(\Phi(\eta), \Phi(\gamma))}{\mathbb{W}_1(\eta, \gamma)} \cdot \sup_{\mu \neq \nu} \frac{\mathbb{W}_1(\Gamma(\mu), \Gamma(\nu))}{\mathbb{W}_1(\mu, \nu)} \\
&= \tau(\Phi)\tau(\Gamma).
\end{aligned}
$$

2. Since $\mathbb{W}_1(\delta_x, \delta_y) = d(x, y)$ and $\delta_x K = K(x, \bullet)$ we have:

$$\sup_{x \neq y} \frac{\mathbb{W}_1(K(x, \bullet), K(y, \bullet))}{d(x, y)} = \sup_{\delta_x \neq \delta_y} \frac{\mathbb{W}_1(\delta_x K, \delta_y K)}{\mathbb{W}_1(\delta_x, \delta_y)} \leq \sup_{\mu \neq \nu} \frac{\mathbb{W}_1(\mu K, \nu K)}{\mathbb{W}_1(\mu, \nu)}.$$

For the reverse inequality,

$$
\begin{aligned}
\mathbb{W}_1(\mu K, \nu K) &= \sup_{f \in Lip(1)} |\mu K(f) - \nu K(f)| \\
&= \sup_{f \in Lip(1)} |\mu(Kf) - \nu(Kf)| \\
&\leq \sup_{f \in Lip(1)} \|Kf\|_{Lip(1)} \cdot \sup_{g \in Lip(1)} |\mu(g) - \nu(g)| \\
&\leq \sup_{f \in Lip(1)} \|Kf\|_{Lip(1)} \cdot \mathbb{W}_1(\mu, \nu)
\end{aligned}
$$

and

$$
\begin{aligned}
\sup_{f \in Lip(1)} \|Kf\|_{Lip(1)} &= \sup_{f \in Lip(1)} \sup_{x \neq y} \frac{\int K(x, dz)f(z) - \int K(y, dz)f(z)}{d(x, y)} \\
&= \sup_{f \in Lip(1)} \sup_{x \neq y} \frac{\int [K(x, dz) - K(y, dz)]f(z)}{d(x, y)} \\
&= \sup_{x \neq y} \frac{\mathbb{W}_1(K(x, \bullet), K(y, \bullet))}{d(x, y)}.
\end{aligned}
$$

Dividing by $\mathbb{W}_1(\mu, \nu)$ gives us the reverse inequality and concludes the proof.

3. By definition, we have:

$$\mathbb{W}_1(\nu K_1, \nu K_2) = \sup_{f \in Lip(1)} |\nu K_1(f) - \nu K_1(f)|$$

$$= \sup_{f \in Lip(1)} \left| \iint \nu(dx) K_1(x, dy) f(y) - \iint \nu(dx) K_2(x, dy) f(y) \right|$$

$$\leq \sup_{f \in Lip(1)} \int \nu(dx) \left| \int K_1(x, dy) f(y) - K_2(x, dy) f(y) \right|$$

$$\leq \int \nu(dx) \mathbb{W}_1(K_1(x, \bullet), K_2(x, \bullet)).$$

**Lemma F.3** *For any $f : \mathbb{R}^d \to \mathbb{R}$, we have*

$$\|f\|_{Lip} = \sup_{x \neq y, \|x-y\| \leq 1} \frac{|f(x) - f(y)|}{\|x - y\|}. \tag{20}$$

**proof**  Let $x \neq y$ and $L := \sup_{x \neq y, \|x-y\| \leq 1} \frac{|f(x)-f(y)|}{\|x-y\|} \leq \infty$. First, assume $\|f\|_{Lip}, L < \infty$. It is clear that $L \leq \|f\|_{Lip}$ since $\{x \neq y, \|x - y\| \leq 1\} \subset \{x \neq y\}$. For the reverse inequality, we split the segment $[x, y]$ into the minimum number of chunks of lengths smaller than 1: $x = z_1 \to z_2 \to \cdots \to z_k = y$ (in particular, if $\|x - y\| \leq 1$ then $z_2 = y$). Then

$$|f(x) - f(y)| \leq \sum_{1 \leq i \leq k-1} |f(z_i) - f(z_{i+1})|$$

$$\leq L \sum_{1 \leq i \leq k-1} \|z_i - z_{i+1}\| = L\|x - y\|.$$

which gives $\|f\|_{Lip} \leq L$ so $L = \|f\|_{Lip}$. Now if $\|f\|_{Lip} = \infty$ but $L < \infty$, by applying the above argument we can obtain a contradiction. Finally, it suffices to note that the case where $\|f\|_{Lip} < \infty$ but $L = \infty$ is impossible since $\|f\|_{Lip} \geq L$.

**Lemma F.4** *For any $n$ and $(z_1, \cdots, z_n) \in \mathbb{R}_+^n$:*

$$f(z_1, \cdots, z_n) := \frac{\sum_{i=1}^n z_i e^{-z_i^2}}{1 + \sum_{i=1}^n e^{-z_i^2}} \leq \sqrt{\ln n + \frac{1}{2e}}. \tag{21}$$

**proof**  $f$ is clearly bounded on $\mathbb{R}_+^n$ ($z_i e^{-z_i^2} \to 0$ when $z_i \to \infty$). Let us now compute the partial derivatives of $f$. For a given $z_i$:

$$\frac{\partial f}{\partial z_i} = \frac{e^{-z_i^2}}{1 + \sum_{k=1}^n e^{-z_k^2}} [1 - 2z_i^2 + 2z_i f(z_1, \cdots, z_n)].$$

There is only one positive solution of $1 - 2z_i^2 + 2z_i f^* = 0$, meaning that $f$ reaches its maximum when all its coordinates are equal. We thus only need to study:

$$g(x) := \frac{nxe^{-x^2}}{1 + ne^{-x^2}} = \frac{xe^{\ln n - x^2}}{1 + e^{\ln n - x^2}}. \tag{22}$$

The change of variable $y = \ln n - x^2$ gives $g(y) = \frac{\sqrt{\ln n - y} e^y}{1 + e^y} \leq \frac{\sqrt{\ln n - y}}{1 + e^{-y}}$ with $y \in ]-\infty, \ln n]$.

On $[0, \ln n]$, we clearly have $g(y) \leq \sqrt{\ln n}$. Let us consider $y \in ]-\infty, 0]$. We get $g^2(y) = \frac{\ln n - y}{(1 + e^{-y})^2} \leq \frac{\ln n - y}{e^{-2y}} \leq \ln n + \frac{1}{2e}$ with since $(2e)^{-1}$ is the maximum of of $ze^{-2z}$ on $\mathbb{R}_+$. This concludes the proof.

**Lemma F.5** *Let $\mu_1, \mu_2, \nu_1, \nu_2 \in_1 (\mathbb{R}^d)$. Then*

$$\mathbb{W}_1(\mu_1 \otimes \mu_2, \nu_1 \otimes \nu_2) \leq \mathbb{W}_1(\mu_1, \nu_1) + \mathbb{W}_1(\mu_2, \nu_2)$$

**proof**  Let $\gamma_1 \in (\mu_1, \nu_1), \gamma_2 \in (\mu_2, \nu_2)$ be optimal for $c(x, y) = \|x - y\|_1$. Note that $\gamma_1 \otimes \gamma_2 \in (\mu_1 \otimes \mu_2, \nu_1 \otimes \nu_2)$, i.e. $\gamma_1 \otimes \gamma_2$ is a transfer plan with the correct marginals, by considering

$$
\int_{X \times X} \mathrm{d}\gamma_1 \otimes \gamma_2(x_1, x_2, y_1, y_y) = \int_{X \times X} \mathrm{d}\gamma_1(x_1, y_1) \mathrm{d}\gamma(x_2, y_2)
$$

$$
= \int_X \mathrm{d}\gamma_1(x_1, y_1) \int_X \mathrm{d}\gamma_2(x_2, y_2)
$$

$$
= \nu_1(\mathrm{d}y_1)\nu_2(\mathrm{d}y_2) = \mathrm{d}\nu_1 \otimes \nu_2(y_1, y_2)
$$

and same for the other marginals.

Thus we have

$$
\mathbb{W}_1(\mu_1 \otimes \mu_2, \nu_1 \otimes \nu_2) = \inf_{\gamma \in (\mu_1 \otimes \mu_2, \nu_1 \otimes \nu_2)} \int \|(x_1, x_2) - (y_1, y_2)\| \mathrm{d}\gamma(x_1, x_2, y_1, y_2)
$$

$$
= \inf_{\gamma \in (\mu_1 \otimes \mu_2, \nu_1 \otimes \nu_2)} \int (\|x_1 - y_1\| + \|y_1, y_2\|) \mathrm{d}\gamma(x_1, x_2, y_1, y_2)
$$

$$
= \inf_{\gamma \in (\mu_1 \otimes \mu_2, \nu_1 \otimes \nu_2)} \int \|x_1 - y_1\| \mathrm{d}\gamma(x_1, x_2, y_1, y_2) + \cdots
$$

$$
\cdots + \inf_{\gamma \in (\mu_1 \otimes \mu_2, \nu_1 \otimes \nu_2)} \int \|x_2 - y_2\| \mathrm{d}\gamma(x_1, x_2, y_1, y_2)
$$

$$
\leq \int \|x_1 - y_1\| \mathrm{d}\gamma_1 \otimes \gamma_2(x_1, x_2, y_1, y_2) + \int \|x_2 - y_2\| \mathrm{d}\gamma_1 \otimes \gamma_2(x_1, x_2, y_1, y_2)
$$

$$
= \int \|x_1 - y_1\| \mathrm{d}\gamma_1(x_1, y_1) + \int \|x_2 - y_2\| \mathrm{d}\gamma_2(x_2, y_2)
$$

$$
= \mathbb{W}_1(\mu_1, \nu_1) + \mathbb{W}_1(\mu_2, \nu_2)
$$

Then using the previous lemma, we prove the following proposition at first.

**Proposition F.6** *Let $E = \mathbb{R}^d$ and suppose $X = \{X_1, \ldots, X_N\}$ and $Y = \{Y_1, \ldots, Y_M\}$. Let $G(x, y) = \exp(-\|x - y\|_2^2)$, $L(x, \mathrm{d}y) = \delta_{l(x)}(\mathrm{d}y)$, and $\Pi$ be the usual projection onto $\mathcal{F}_\delta$. Then for $\mu = m(X)$ and $\nu = m(Y)$,*

$$
\mathbb{W}_1(\mu A_\mu, \nu A_\nu) \leq 2\tau(\Pi)\tau(L) \left[ \sqrt{d}\sqrt{\ln(\min(N, M)) + \frac{1}{2e}}\|G\|_{Lip} + \|G\|_\infty + \sqrt{d} + 2 \right] \mathbb{W}_1(\mu, \nu).
$$

**proof**  We use the Kantorovich formulation of $\mathbb{W}_1$. Let $f$ be a function with $\|f\|_{Lip} \leq 1$. We can assume without loss of generality that $f(y) = 0$. For simplicity, we write $G(x, \bullet) = G_x$. We wish to upper-bound the quantity $|\Psi_{G_x}(\mu)(f) - \Psi_{G_y}(\nu)(f)|$.

Because $\Psi_{G_x}$ and $\Psi_{G_y}$ are homonegeous in their measure argument, and for the sake of simplicity, we write $\mu = \sum_i \delta_{x_i}$ $\nu = \sum_i \delta_{y_i}$ (which is equivalent to simplifying by $1/N$ in e.g. the numerator and denominator of $\Psi_{G_x}$). This guarantees in particular that $\mu(G_x) \geq 1$ and $\nu(G_y) \geq 1$ ($x$ and $y$ are in $\mu$ and $\nu$ resp.) and equivalently that $1/\mu(G_x) \leq 1$ and $1/\nu(G_y) \leq 1$.

Then:

$$
|\Psi_{G_x}(\mu)(f) - \Psi_{G_y}(\nu)(f)| = \frac{1}{\mu(G_x)\nu(G_y)} |\nu(G_y)\mu(G_x f) - \mu(G_x)\nu(G_y f)|
$$

$$
= \frac{1}{\mu(G_x)\nu(G_y)} |\nu(G_y)\mu(G_x f) - \nu(G_y)\nu(G_y f) + \nu(G_y)\nu(G_y f) - \mu(G_x)\nu(G_y f)|
$$

$$
\leq \frac{\nu(G_y)}{\mu(G_x)\nu(G_y)} |\mu(G_x f) - \nu(G_y f)| + \frac{\nu(G_y f)}{\mu(G_x)\nu(G_y)} |\nu(G_y) - \mu(G_x)|.
$$

$$
\tag{23}
$$

We start by bounding the second term of equation 23. We have:

$$\frac{\nu(G_y f)}{\mu(G_x)\nu(G_y)}|\nu(G_y) - \mu(G_x)| = \frac{\nu(G_y f)}{\mu(G_x)\nu(G_y)}|(\delta_x \otimes \mu)(G) - (\delta_y \otimes \nu)(G)|$$

$$\leq \frac{\nu(G_y f)}{\mu(G_x)\nu(G_y)}\|G\|_{Lip}\mathbb{W}_1(\delta_x \otimes \mu, \delta_y \otimes \nu).$$

Here, $\delta_x \otimes \mu$ denotes the product of the two measures on $E \times E$. Since $f(y) = 0$, we see that $f(z) \leq f(y) + \|f\|_{Lip}\|y - z\|_1 \leq \|y - z\|_1$. This gives:

$$\frac{\nu(G_y f)}{\nu(G_y)} = \frac{\int G_y(z)f(z)\nu(\mathrm{d}z)}{\int G_y(z)\nu(\mathrm{d}z)} \leq \frac{\int G_y(z)\|y - z\|_1\nu(\mathrm{d}z)}{\int G_y(z)\nu(\mathrm{d}z)}$$

$$\leq \frac{\sum_{i=1}^N G(y, y_i)\|y - y_i\|_1}{\sum_{i=1}^N G(y, y_i)} \leq \sqrt{d}\frac{\sum_{i=1}^N e^{-\|y-y_i\|_2^2}\|y - y_i\|_2}{\sum_{i=1}^N e^{-\|y-y_i\|_2^2}},$$

where we applied Cauchy-Schwartz for the last inequality. Since $y = y_i$ for a given $i$, we are interested in the quantity $\frac{\sum_{i=1}^{N-1} z_i e^{-z_i^2}}{1 + \sum_{i=1}^{N-1} e^{-z_i^2}}$ for arbitrary $z_i \geq 0$. Applying Lemma F.4 with $n = N - 1$ gives an upper-bound of $\sqrt{\ln N + \frac{1}{2e}}$.

Let us now consider the first term of equation 23:

$$\frac{\nu(G_y)}{\mu(G_x)\nu(G_y)}|\mu(G_x f) - \nu(G_y f)| = \frac{1}{\mu(G_x)}|\mu(G_x f) - \nu(G_y f)|$$

$$\leq \frac{1}{\mu(G_x)}\|Gf\|_{Lip}\mathbb{W}_1(\delta_x \otimes \mu, \delta_y \otimes \nu).$$

To estimate $\|Gf\|_{Lip}$ we have

$$\|Gf\|_{Lip} = \sup_{(x,w)\neq(y,z)} \frac{|G(x,w)f(w) - G(y,z)f(z)|}{\|(x,w) - (y,z)\|_1}$$

where additionally, we can assume that $\|(x,w) - (y,z)\| \leq 1$ (see Lemma F.3). We have:

$$|G(x,w)f(w) - G(y,z)f(z)| = |G(x,w)f(w) - G(x,w)f(z) + G(x,w)f(z) - G(y,z)f(z)|$$

$$\leq |G(x,w)||f(w) - f(z)| + |f(z)||G(x,w) - G(y,z)|.$$

For the first term, we see that

$$|G(x,w)||f(w) - f(z)| \leq \|G\|_{\infty,\infty}\|f\|_{Lip}d(w,z)$$

$$\leq \|G\|_{\infty,\infty}\|f\|_{Lip}(d(w,z) + d(x,y)).$$

For the second term, we have

$$|f(z)||G(x,w) - G(y,z)| \leq \|y - z\|_1|G(x,w) - G(y,z)|$$

$$\leq \|y - z\|_1\|\nabla G(t_1, t_2))\|_\infty\|(x,w) - (y,z)\|_1,$$

for $t_1$ in the segment $[x,y]$ and $t_2$ in the segment $[w,z]$ (this follows directly from the mean value theorem, note that the gradient is taken with respect to both variables). We used $f(y) = 0$ and $f(z) \leq f(y) + \|f\|_{Lip}\|y - z\|_1 = \|y - z\|_1$ in the first line.

In the Gaussian case:

$$\|y - z\|_1\|\nabla G(t_1, t_2))\|_\infty \leq (\|y - t_1\|_1 + \|t_1 - t_2\|_1 + \|t_2 - z\|_1)2\|t_1 - t_2\|_\infty e^{-\|t_1-t_2\|_2^2}$$

$$\leq 2(2 + \|t_1 - t_2\|_1)\|t_1 - t_2\|_\infty e^{-\|t_1-t_2\|_2^2},$$

where we used the fact that $\|y - t_1\|_1 \leq 1$ and $\|t_2 - z\|_1 \leq 1$ ($t_1$ is in the $[x,y]$ segment and $\|x - y\|_1 \leq 1$ by assumption). That upper bound is uniformly bounded with respect to $t_1$ and $t_2$, we let $C$ denote that constant. A loose upper-bound on $C$ is $\sqrt{d} + 2$ (which we use in the statement of the proposition).

To conclude, it suffices to note that by Lemma F.5 we have

$$\mathbb{W}_1(\delta_x \otimes \mu, \delta_y \otimes \nu) \leq \mathbb{W}_1(\delta_x, \delta_y) + \mathbb{W}_1(\mu, \nu).$$

Finally, we can prove the theorem based on the previous lemma and proposition.

**Proof.** Firstly, using Proposition 4.2, we know that $\mu A_\mu$ is another empirical measure concentrated on $\{\mathrm{Attention}(x_i, X, X)\}$, similarly, $\nu A_\nu$ is concentrated on $\{\mathrm{Attention}(y_i, Y, Y)\}$. This fact allows us to use the following result from Santambrogio (2015) Equation 6.2

$$\mathbb{W}_1(\mu, \nu) = \min\left\{\sum_{i,j} \gamma_{ij} d(x_i, y_j)\gamma_{i,j} \geq 0, \ \sum_i \gamma_{ij} = \frac{1}{M}, \ \sum_j \gamma_{ij} = \frac{1}{N}\right\},$$

Applied to $\mathbb{W}_1(\mu A_\mu, \nu A_\nu)$, it gives

$$\mathbb{W}_1(\mu A_\mu, \nu A_\nu) = \min\Big\{\sum_{i,j} \gamma_{ij} d(\mathrm{Attention}(x_i, X, X), \mathrm{Attention}(y_j, Y, Y))$$

$$\gamma_{i,j} \geq 0, \ \sum_i \gamma_{ij} = \frac{1}{M}, \ \sum_j \gamma_{ij} = \frac{1}{N}\Big\}$$

$$= \min\Big\{\sum_{i,j} \gamma_{ij} \mathbb{W}_1(A_\mu(x_i, \bullet), A_\nu(y_i, \bullet))$$

$$\gamma_{i,j} \geq 0, \ \sum_i \gamma_{ij} = \frac{1}{M}, \ \sum_j \gamma_{ij} = \frac{1}{N}\Big\}.$$

Using Lemma F.2 for each term, we have

$$\mathbb{W}_1(A_\mu(x_i, \bullet), A_\nu(y_j, \bullet)) \leq \tau(\Pi)\tau(L)\mathbb{W}_1(\Psi_{G(x_i, \bullet)}(\mu), \Psi_{G(y_j, \bullet)}(\nu)).$$

Now, from Proposition F.6 ($x_i$ belongs to $\mu$ and $y_j$ to $\nu$), we get

$$\mathbb{W}_1(\Psi_{G(x_i, \bullet)}(\mu), \Psi_{G(y_j, \bullet)}(\nu))$$
$$\leq \left[\sqrt{d}\sqrt{\ln N + \frac{1}{2e}}\|G\|_{Lip} + \|G\|_\infty + \sqrt{d} + 2\right](d(x_i, y_j) + \mathbb{W}_1(\mu, \nu)).$$

Substituting this back into the above formula, we obtain

$$\mathbb{W}_1(\mu A_\mu, \nu A_\nu)$$

$$\leq \min\Big\{\sum_{i,j} \gamma_{ij}\mathbb{W}_1(A_\mu(x_i, \bullet), A_\nu(y_i, \bullet))\gamma_{i,j} \geq 0, \ \sum_i \gamma_{ij} = \frac{1}{M}, \ \sum_j \gamma_{ij} = \frac{1}{N}\Big\}$$

$$\leq \tau(\Pi)\tau(L)\min\Big\{\sum_{i,j} \gamma_{ij}\left[\sqrt{d}\sqrt{\ln N + \frac{1}{2e}}\|G\|_{Lip} + \|G\|_\infty + \sqrt{d} + 2\right](d(x_i, y_j) + \mathbb{W}_1(\mu, \nu))$$

$$\gamma_{i,j} \geq 0, \ \sum_i \gamma_{ij} = \frac{1}{M}, \ \sum_j \gamma_{ij} = \frac{1}{N}\Big\}$$

$$= \tau(\Pi)\tau(L)\left[\sqrt{d}\sqrt{\ln N + \frac{1}{2e}}\|G\|_{Lip} + \|G\|_\infty + \sqrt{d} + 2\right]\Big(\mathbb{W}_1(\mu, \nu) +$$

$$\min\Big\{\sum_{i,j} \gamma_{ij}d(x_i, y_j)\gamma_{i,j} \geq 0, \ \sum_i \gamma_{ij} = \frac{1}{M}, \ \sum_j \gamma_{ij} = \frac{1}{N}\Big\}\Big)$$

$$= \tau(\Pi)\tau(L)\left[\sqrt{d}\sqrt{\ln N + \frac{1}{2e}}\|G\|_{Lip} + \|G\|_\infty + \sqrt{d} + 2\right](\mathbb{W}_1(\mu, \nu) + \mathbb{W}_1(\mu, \nu))$$

$$= 2\tau(\Pi)\tau(L)\left[\sqrt{d}\sqrt{\ln N + \frac{1}{2e}}\|G\|_{Lip} + \|G\|_\infty + \sqrt{d} + 2\right]\mathbb{W}_1(\mu, \nu),$$

where we used in particular $\sum_{i,j} \gamma_{ij} = 1$. The inequality being valid for both $M$ and $N$, taking the min gives the result. Let $m(X_M^{(t+1)}) = \mu A_\mu$ and $m(X_N^{(t+1)}) = \nu A_\nu$, then the Theorem 4.6 proof is finished.

Table 3: Perplexity (PPL) and Accuracy (ACC) on language modeling with evaluation lengths from IID (256) to OOD (4k). We train the NanoGPT-124M on the OpenWebText2 from scratch.

| Methods | Train Length | Evaluate Length | | | | | | | | | |
|---|---|---|---|---|---|---|---|---|---|---|---|
| | | 256 | | 512 | | 1024 | | 2048 | | 4096 | |
| | | PPL↓ | ACC↑ | PPL↓ | ACC↑ | PPL↓ | ACC↑ | PPL↓ | ACC↑ | PPL↓ | ACC↑ |
| NoPE | | | | | | | | | | | |
| NanoGPT | 256 | 30.15 | 38.33 | 44.10 | 34.45 | 369.15 | 20.47 | >1000 | 10.43 | >1000 | 5.30 |
| NanoGPT | 512 | 28.42 | 38.33 | 27.15 | 39.04 | 51.12 | 33.14 | 531.02 | 17.62 | >1000 | 9.51 |
| *detach* | 256+256 | 29.83 | 38.44 | 29.24 | 38.26 | 60.09 | 31.77 | 832.53 | 16.62 | >1000 | 8.60 |
| *random* | 256+256 | 30.58 | 38.14 | 33.11 | 36.93 | 72.92 | 29.02 | 148.33 | 22.51 | 240.41 | 19.54 |
| RoPE | | | | | | | | | | | |
| NanoGPT | 256 | 28.87 | 38.81 | 44.22 | 34.09 | 118.31 | 24.60 | 272.91 | 17.95 | 534.82 | 14.18 |
| NanoGPT | 512 | 26.90 | 39.59 | 25.66 | 39.81 | 52.61 | 32.69 | 172.67 | 22.80 | 425.46 | 17.39 |
| *detach* | 256+256 | 42.03 | 35.25 | 39.73 | 35.73 | 49.49 | 33.27 | 101.26 | 25.37 | 172.44 | 20.62 |
| *random* | 256+256 | 42.89 | 35.44 | 41.61 | 35.48 | 63.62 | 30.92 | 131.61 | 23.46 | 213.40 | 19.48 |

# G  USEFUL AND FAILURE CASES OF THEOREM 4.6

**Successful case: similiar semantic meanings.**  When $\mu$ and $\nu$ are sampled from the same underlying distribution $\mathcal{P}'$, that is to say: $\lim_{N \to \infty} m(X) = \lim_{M \to \infty} m(Y)$. In this case, $\mathbb{W}_1(\mu, \nu)$ can be bounded to the constant $C(\mathcal{P}')$ even as the test length $M$ increases and thus the distance $\mathbb{W}_1\left(\mu \hat{A}_\mu^N, \nu \hat{A}_\nu^N\right)$ is fixed with the length term $\sqrt{\ln N}$, resulting that the pure self-attention can generalize to out-of-distribution length sentences. This observation is consistent with the findings in Bhattamishra et al. (2024) suggesting that transformer decoders can easily copy long strings and Liu et al. (2022) showing that transformers can in theory simulate many finite-state automata in principle. The consistency in these behaviors arises because both long and short sequences follow identical underlying rules for such tasks, allowing sequences of varying lengths to be modeled within the same distributional framework.

**Failure case: diverge semantic meanings and minor permutation.**  Our theoretical analysis of length generalization is fundamentally driven by the relationship between word embeddings and sentence-level semantic meanings. We identify two critical scenarios where self-attention mechanisms may fail to generalize effectively: (1) If $\mathbb{W}_1(\mu, \nu)$ diverges, the upper bound established in Theorem 4.6 grows significantly, making our theoretical bound uninformative. This weakens the constraint and ultimately reduces the length generalization capability. (2) When the geometry of the embeddings aligns closely while the semantic meanings of the two sentences differ significantly, it demonstrates a capacity for ineffective generalization. For example, the transformers may fail to distinguish the pair of negated sentences (Singh et al., 2023; Zhang et al., 2023) because word embeddings are similar while the semantic meaning of the whole sentence is quite different. This observation aligns with findings in Zhou et al. (2024); Huang et al. (2024).

# H  EXPERIMENTS

## H.1  PRE-TRAINING EXPERIMENTS

Given a token matrix $X$, a language model $\theta$ is trained to maximize the conditional likelihood $P_\theta(X_i \mid X_{<i}), i \in [N]$. Standard fine-tuning minimizes the loss over all tokens:

$$L = -\frac{1}{M} \sum_{i=1}^{M} \log P_\theta(X_i \mid X_{<i}),  \tag{24}$$

which adapts the model to the entire long sequence.

To show that *VCL* is effective in the pre-training phase, we train NanoGPT-124M from scratch using our methods. The training process comprises 6000 steps, employing a global batch size of 24 and gradient accumulation steps of $5 \times 8$ on 8 A100 GPUs. We use a learning rate $6e-4$ and a weight decay $1e-1$, with 2000 warm-up steps. We use AdamW optimizer with its default hyperparameters setup. Since the default positional encoding from NanoGPT is the absolute PE with no extrapolation

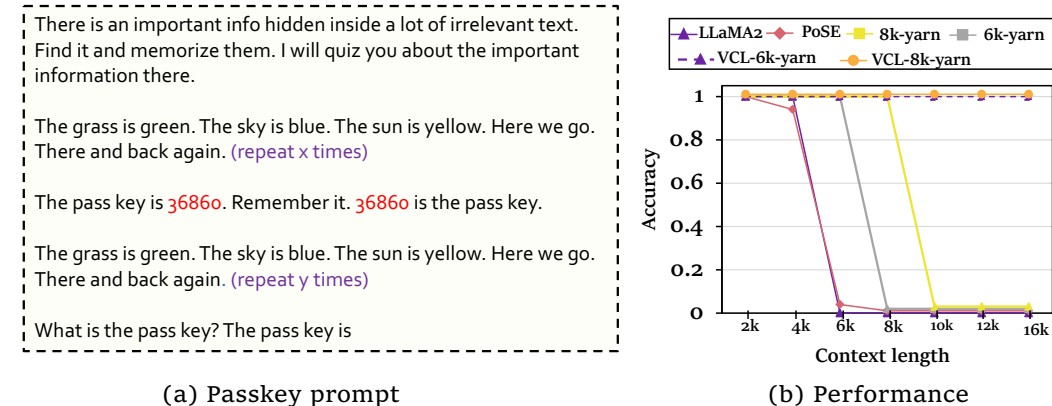

(a) Passkey prompt

(b) Performance

Figure 7: (a) Prompt template used for passkey retrieval; (b) Retrieval accuracy for *VCL* compared with other baselines. 6k-yarn and 8k-yarn stand fine-tuning on 6k and 8k context windows with yarn. *VCL*-6k-yarn and *VCL*-8k-yarn stand fine-tuning on 6k and 8k context windows with *VCL* integrating yarn.

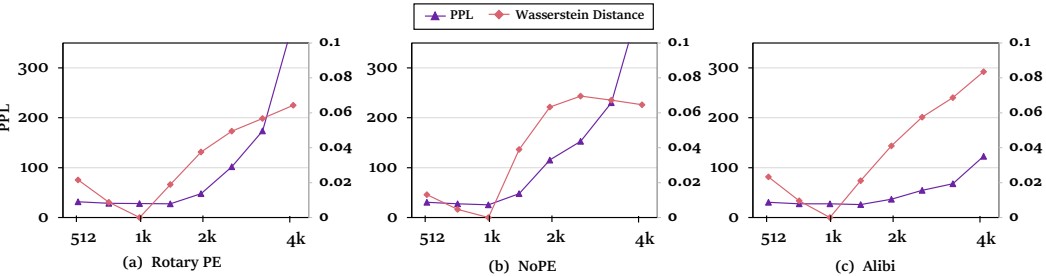

(a) Rotary PE

(b) NoPE

(c) Alibi

Figure 8: PPL v.s. Wasserstein Distance on different lengths. We compare CodeLLM-1B (Kazemnejad et al., 2023) pre-training on the fixed context window 1024 under the same settings on different positional encodings: RoPE (Su et al., 2022), NoPE (Kazemnejad et al., 2023), and Alibi (Press et al., 2022). The left y-axis is for PPL and the right y-axis is for Wasserstein Distance. The x-axis is for different context window sizes ranging from 512 to 4096.

ability, we adopt the NoPE and RoPE for the default settings. The pretraining dataset is sourced from the OpenWebText2 (Gao et al., 2020), with a train block size ranging from 256 to 512 as the baseline and oracle respectively. We use the traditional loss function and our *VCL* loss to pre-train the NanoGPT from scratch as following:

$$L_{pre-train} = \alpha L + (1 - \alpha)L_{VCL} \tag{25}$$

We select $\alpha$ from $\{0.1, 0.5, 0.9\}$ and observed that the smaller $\alpha$ induced a better performance of the length extrapolation.

*detach* means that we fix k=256 with the real tokens and the whole input sequence is M =512. *random* means that we fix k=256 with the randomly generated tokens with the Gaussian distribution of the latter 256 tokens' mean and variance while the whole input sequence is M =512. Since all of the PPLs are very close to each other (around 1.0), we did not divide the loss by the length to better distinguish them among different methods. Table 3 shows that *VCL* is on par with or even better than the oracle under the NoPE settings by only adding one actual token, especially in the out-of-distribution length (more than 1024). All three methods on NanoGPT outperform the initial pertaining models, verifying the superiority of our insight that alleviates the length term while enhancing the ability of length generalization simultaneously. Furthermore, we extend our *VCL* pertaining methods to the RoPE setting.

Table 4: Accuracy on LongBench with extreme length (more than 8k).

| Methods | Single-Doc QA | Multi-Doc QA | Summarization | Few-shot Learning | Synthetic Task | Code Completion | Avg |
|---|---|---|---|---|---|---|---|
| | | | llama2-7b | | | | |
| Original | 4.12 | 3.19 | 9.54 | 65.32 | 1.00 | 58.00 | 26.74 |
| Full-length (8k) | 9.33 | 7.62 | 15.11 | 62.47 | 3.08 | 56.94 | 28.58 |
| RandPE | 9.79 | 7.92 | 17.01 | 58.96 | 5.46 | 56.88 | 28.54 |
| PoSE | 11.85 | 8.38 | 16.91 | 62.53 | 4.07 | 53.01 | 28.93 |
| NTK-aware | 11.39 | 7.47 | 15.79 | 61.53 | 4.01 | 56.75 | 28.88 |
| Yarn | 8.96 | 7.87 | 16.77 | 60.33 | 3.72 | 57.55 | 28.52 |
| *VCL*-Linear-1k | 8.23 | 8.73 | 22.16 | 61.91 | 4.50 | 60.02 | 30.23 |
| *VCL*-NTK-1k | 10.73 | 8.77 | 17.63 | 61.83 | 4.96 | 59.20 | 29.85 |
| *VCL*-Yarn-1k | 13.20 | 8.15 | 16.87 | 62.94 | 6.12 | 55.73 | 29.92 |
| | | | llama2-7b-chat-4k | | | | |
| Original | 24.9 | 22.6 | 24.7 | 60.0 | 5.9 | 48.1 | 31.0 |
| *VCL*-Linear-1k | 13.86 | 26.64 | 24.44 | 62.47 | 7.38 | 54.64 | 33.95 |
| *VCL*-NTK-1k | 12.61 | 25.19 | 24.95 | 62.25 | 4.96 | 56.27 | 33.44 |
| *VCL*-Yarn-1k | 19.52 | 26.20 | 25.38 | 62.74 | 4.75 | 53.66 | 34.41 |

## H.2 MORE LLMs EXPERIMENTS

We also trained our methods *VCL* on the llama2-chat model as seen in Table 4.

## H.3 METRICS ON CODELLM-1B

Here we give the distribution metrics and the ppl on different contexts window size across RoPE, NoPE, and Alibi. We demonstrate the Wasserstein Distance, JS Distance, and KL Divergence, as well as the PPL in the following tables.

Table 5: Performance Metrics for RoPE

| Context Window Size | PPL | Wasserstein Distance | JS Distance | KL Divergence |
|---|---|---|---|---|
| 512 | 30.625 | 0.0132 | 0.0256 | 0.0026 |
| 760 | 27.500 | 0.0047 | 0.0162 | 0.0010 |
| 1024 | 25.750 | 0.0000 | 0.0000 | 0.0000 |
| 1600 | 48.250 | 0.0391 | 0.1214 | 0.0630 |
| 2048 | 115.500 | 0.0633 | 0.1981 | 0.1705 |
| 2560 | 153.000 | 0.0695 | 0.2378 | 0.2429 |
| 3072 | 230.000 | 0.0673 | 0.2531 | 0.2725 |
| 4096 | 444.000 | 0.0646 | 0.2660 | 0.3063 |

Table 6: Performance Metrics for NoPE

| Context Window Size | PPL | Wasserstein Distance | JS Distance | KL Divergence |
|---|---|---|---|---|
| 512 | 32.000 | 0.0216 | 0.0357 | 0.0051 |
| 760 | 28.750 | 0.0088 | 0.0213 | 0.0018 |
| 1024 | 28.375 | 0.0000 | 0.0000 | 0.0000 |
| 1600 | 27.875 | 0.0190 | 0.0350 | 0.0050 |
| 2048 | 48.250 | 0.0376 | 0.0722 | 0.0213 |
| 2560 | 102.000 | 0.0495 | 0.1021 | 0.0428 |
| 3072 | 174.000 | 0.0567 | 0.1237 | 0.0629 |
| 4096 | 392.000 | 0.0643 | 0.1536 | 0.0970 |

Table 7: Performance Metrics for Alibi

| Context Window Size | PPL | Wasserstein Distance | JS Distance | KL Divergence |
|---|---|---|---|---|
| 512 | 30.625 | 0.0234 | 0.0452 | 0.0082 |
| 760 | 27.875 | 0.0097 | 0.0305 | 0.0037 |
| 1024 | 27.875 | 0.0000 | 0.0000 | 0.0000 |
| 1600 | 26.250 | 0.0212 | 0.0387 | 0.0060 |
| 2048 | 37.000 | 0.0410 | 0.0593 | 0.0139 |
| 2560 | 54.500 | 0.0575 | 0.0787 | 0.0242 |
| 3072 | 68.000 | 0.0687 | 0.0935 | 0.0337 |
| 4096 | 123.000 | 0.0835 | 0.1169 | 0.0515 |

