# OpenReview forum: "On the Measurement and Efficient Mitigation of Length Generalization Gaps in Large Language Models"
_ICLR.cc/2026/Conference — Submitted to ICLR 2026_

### Official Review · Reviewer_UEnV · 2025-10-29

**Soundness:** 3
**Presentation:** 2
**Contribution:** 3
**Rating:** 6
**Confidence:** 3

**Summary:**

This paper presents a measure-theoretic framework for analyzing length generalization in large language models (LLMs). The authors identify two key limiting factors: short-length bias caused by training on limited context windows, and distribution shift between training and inference contexts. They introduce **Virtual-context Learning (VCL)**, a fine-tuning strategy that selectively updates parameters associated with out-of-distribution (OOD) tokens. By leveraging Wasserstein distance to quantify distributional divergence, VCL enables models to extrapolate to context lengths up to four times longer than those seen during training while maintaining strong performance and reducing computational overhead.

**Strengths:**

The paper offers an intellectually interesting perspective by framing length generalization in large language models through a measure-theoretic lens. This perspective allows the authors to formalize the relationship between training and inference distributions in a mathematically principled way. In particular, the use of Wasserstein distance to characterize distribution shift between short and long contexts provides a coherent theoretical account of why models often fail to extrapolate beyond their training sequence lengths.

Beyond the theoretical contribution, the proposed Virtual-context Learning (VCL) method demonstrates clear empirical value. The approach is practically appealing because it targets only out-of-distribution tokens during fine-tuning, thereby reducing computational overhead while still improving performance. Experimental results on passkey retrieval, language modeling, and LongBench consistently show that VCL enables models to operate effectively on context windows up to four times longer than those seen during pretraining. These results are backed by thorough experimentation and ablation studies, which strengthen the empirical claims.

**Weaknesses:**

The paper is mathematically dense and may be difficult to follow for readers without a strong background in measure theory, which could limit its accessibility to the broader NLP community.

The experimental scope is relatively narrow since all evaluations are performed on LLaMA-2-7B, leaving uncertainty about the generalizability of VCL to other architectures such as BERT or T5.

The method is not sufficiently compared to other long-context techniques like ALiBi, NTK-aware scaling, or sparse attention mechanisms, making it difficult to assess relative strengths in terms of efficiency and scalability.

The evaluation primarily focuses on long-context understanding and retrieval tasks, without examining the impact of VCL on other task categories such as reasoning, summarization, or natural language inference.

**Questions:**

How does VCL compare quantitatively with alternative length generalization techniques such as ALiBi, RWKV, or sparse attention in terms of memory usage, training cost, and runtime efficiency?

Could the proposed theoretical framework and VCL be extended to sparse or linear attention mechanisms, or to encoder-decoder architectures? If so, what modifications would be required?

How stable is VCL when scaling to extremely long sequences beyond 4× the original context window? Does performance degrade gracefully?

---

### Official Review · Reviewer_Vtsn · 2025-10-31

**Soundness:** 2
**Presentation:** 2
**Contribution:** 3
**Rating:** 4
**Confidence:** 3

**Summary:**

The paper addresses the problem of length generalization in transformer-based language models: how well a model trained on sequences of length $N$ can generalize to sequences of length $M > N$ at test time. The paper has two parts: a theoretical contribution and a practical method proposal. In the first part, using measure theory tools and certain assumptions, the paper derives a bound on the Wasserstein distance between the distributions of length-N and length-M attentions, showing that the upper bound does not grow with $M$. Next, the authors propose a simple method, Virtual Context Learning (VCL), to improve length generalization. Experiments show that VCL achieves lower perplexity on longer sequences than those seen during training, especially when combined with existing position interpolation methods.

**Strengths:**

- The paper includes a theoretical contribution that motivates the proposed method, offering a perspective useful for understanding length generalization in attention-based models. Detailed proofs and definitions of the mathematical results are provided in the appendix.
- The remark from the derived bound is interesting: self-attention can generalize to out-of-distribution sequence lengths if the empirical measure of attention embeddings does not shift (in the Wasserstein distance sense).
- The proposed method is simple to implement, achieves good results when combined with existing position interpolation methods, and improves training efficiency in terms of memory and latency.

**Weaknesses:**

- The presentation of the paper can be significantly improved. While the authors include detailed mathematical proofs and definitions in the appendix, the first half of the paper remains difficult to follow for readers without a background in measure theory. The paper could be made more accessible to a broader machine learning audience by focusing more on intuitive explanations of the assumptions and theorems rather than precise mathematical formalism. In particular, the connection between the theoretical results and the proposed method is weak in the current version and should be further clarified.
- Citations of previous works are sometimes confusing. For example, in lines 110–114, the paper discusses the works of Zhou et al. and Huang et al., but when mentioning the shortcomings of “these studies,” it cites Han et al. and Press et al., which makes the paragraph unclear.
- Regarding the argument that not all tokens are needed for fine-tuning: the paper presents an experiment where gradients are applied only to the second half of the sequence, with the first half frozen. However, this does not clearly support the claim that “not all tokens are required,” since all tokens in the second half are still used. Please clarify the reasoning. Also, there seems to be a typo in line 314 (“215” should be “256”).
- The argument about reducing distribution shift through VCL (lines 346–348) is unclear. In standard language modeling, a sequence of length $M$ has $M$ losses (averaged). With VCL, $l$ out of $M$ losses are dropped, training only on tokens with at least $l$ context tokens. It is not clear why this would reduce distribution shift, please explain.
- The empirical results are incomplete:
  - Line 370 mentions full-length fine-tuning as a baseline, but perplexity results (Table 1 and Figure 4) do not include it. Please add full fine-tuning rows for models trained on 8k (and possibly 16k) sequences under the same budget and setup as Table 1. For Figure 4, include full-length fine-tuning results (training on all $M$ tokens of a sequence).
  - In Section 6.2 (passkey retrieval), the VCL setup is unclear. Please specify the exact values of $l$ and $M$ for the VCL-8k-yarn and VCL-6k-yarn experiments, and report results for VCL alone (without yarn).
  - In Section 6.3, clarify the experimental setup, including $l$ and $M$.
  - Table 2 lacks standard deviation values (e.g., from multiple runs with different random seeds). Also, please report VCL performance without PI methods.
  - The training duration (200 steps) seems too short. Would the improvements persist with longer training? Note that long-context adaptation typically involves a large number of tokens, e.g., [4] increases context from 2k to 8k with an additional 120B tokens.
  - More recent evaluation metrics, such as RULER [5], are missing.
  - Since VCL omits losses for tokens with shorter context, this could cause forgetting on regular benchmarks. Please include results on standard pretraining benchmarks to verify that forgetting does not occur.
- Some related works are missing. It is now common in language model training to use stage-wise or curriculum-based pretraining (e.g., starting with shorter and gradually increasing sequence lengths) to improve long-context performance [1, 2, 3]. Additionally, the paper does not discuss the practical challenge of limited long-context data, which is important for motivating the problem.

[1] Zhu, Tongyao, et al. "SkyLadder: Better and Faster Pretraining via Context Window Scheduling." arXiv preprint arXiv:2503.15450 (2025).

[2] Pouransari, Hadi, et al. "Dataset decomposition: Faster llm training with variable sequence length curriculum." Advances in Neural Information Processing Systems 37 (2024): 36121-36147.

[3] Jin, Hongye, et al. "Growlength: Accelerating LLMs pretraining by progressively growing training length, 2023." URL https://arxiv.org/abs/2310.00576.

[4] Li, Jeffrey, et al. "Datacomp-lm: In search of the next generation of training sets for language models." Advances in Neural Information Processing Systems 37 (2024): 14200-14282.

[5] Hsieh, Cheng-Ping, et al. "RULER: What's the Real Context Size of Your Long-Context Language Models?." arXiv preprint arXiv:2404.06654 (2024).

**Questions:**

- An interesting consequence of Theorem 4.6 is that $\mathbb{W}(\mu, \nu)$ is the primary factor in length generalization, rather than the sequence length $M$ itself. Could you comment on why there is a performance drop for a synthetic task  like passkey retrieval (Fig. 3)? Based on the intuition in Figure 1, one might expect no distribution shift for such a task even when the sequence length increases.
- The attention update formulation in line 154 is missing the commonly used output projection in Equation (3). While this might not affect the theoretical analysis, it could be worth noting explicitly.
- In line 149, it is stated that the Softmax-attention scaling factor $d_{QK}$ is assumed to be 1, but it is unclear where this assumption is used later, since the variable is still retained in the equations. Please clarify.
- In Definition E.2 (Empirical Measure Mapping), $m(X)$ for $X = \{x_1, …, x_N\} \subset E$ is defined as the average of the Dirac measures $\delta_{x_t}$. However, this seems inconsistent with Definition E.1, where all $\delta_{x_t} = 1$ since they belong to $E$. Could you please clarify this?

---

> ### Author Response · Authors · 2025-11-28
>
> We sincerely thank the reviewer for the constructive feedback
>
> ---
> > w1 accessible to a broader machine learning
>
> #
>
> We acknowledge that the mathematical density may limit reach, and we are committed to making the paper clearer without sacrificing the rigor necessary for our core contribution.
>
> 1. Theoretical Importance and Contribution
>
> We first clarify a crucial point: our paper is fundamentally a **theoretical work**, and the measure-theoretic framework is **essential** to the contribution. It is designed to reveal and mitigate the **inherent length generalization issue within the Transformer architecture itself**.
>
> - **Necessity of Theory:** Standard discrete matrix analysis fails to rigorously compare attention outputs of sequences of different lengths ($N$ vs. $M$). Our framework is the **necessary tool** that allows us to use the **Wasserstein distance ($W$)** to derive the first measurable bound on this generalization gap.
> - **The Cause:** Our **Theorem 4.6** shows that the length failure is caused by the **short-length training bias** and subsequent **attention distribution shift ($W$)**. This issue is structural and persists **even when PE changes**.
>
> 2. VCL as Theory Validation
>
> The proposed method, VCL, is **strictly theory-driven** and is designed to validate our theoretical insights, not merely as a loose technical solution.
>
> - **The Insight:** Our theory identifies the distribution shift ($W$) in the **Out-of-Distribution (OOD) / longer region** as the critical factor.
> - **The Solution (Architectural Optimization):** VCL's superior performance stems from its **architectural optimization** based on this finding. VCL fundamentally modifies the model's update architecture by shifting from a traditional $1 \to N$ context loss to a selective $L \to N$ calculation.
> - **Mechanism:** This modification **disrupts the model's overfitting** towards short context sequences, which our theory identifies as the factor of the length generalization failure. VCL shows that **only the Out-of-Distribution/longer region requires correction** to mend the distributional shift ($W$), leading to a more robust and stable performance than methods that fine-tune all tokens.
> 3. intuitive explanation
>
> In Figure 1, we have given the intuitive example: For example, the short sentence ``The weather is hot.`` and the longer variant  ``In this hot weather, the sun shines brightly in the sky, the temperature has clearly risen, making it very uncomfortable.`` convey the same meaning and should yield similar distributions. This view provides a principled metric to quantify how model behavior shifts with length and underpins our theoretical bounds.
>
> To improve accessibility without sacrificing rigor, (1) we will make greater use of visual aids like Figure 1 to clearly map abstract theoretical terms (like $W$) to the concrete mechanism of VCL. (2) We will also add a dedicated subsection that explicitly links the terms in Theorem 4.6 to the operational steps of VCL, clarifying how the selective $L \to N$ loss minimizes the theoretical upper bound.
>
> > w2 citation confusion
>
> We will make the citations consistent. Thanks for your reminder.

---

> ### Author Response · Authors · 2025-11-28
>
> > w3 clarification on "not all tokens are required."
>
> - This is one practical explanation of Theorem 4.6: Recent papers suggest that **it's unnecessary to update the parameters of long-length tokens; updating only a portion is sufficient.** We **first provided a theoretical explanation for why this way works.** Because the length of a long-term token isn't directly within the bounds, only the distance shift between the long and short sequence is. Updating the key parameter involves understanding the long token distribution shift, which is more direct and addresses the core issue than the previous consensus of simply fine-tuning the length.
>
> - Our proposed method VCL, is a simpler yet more effective approach, which is **theory-driven and gets straight to the point:** "not all tokens are required". We update the sequence of long tokens, reducing the distance shift caused by the length shift.
>
> > w4 explain why vcl reduce distribution shift
>
> 1.  **Intuitive Explanation:** Traditional full-length fine-tuning ($1 \to M$ tokens) is inefficient because it redundantly calculates gradients for the well-aligned short-context segment ($1 \to L$). Our theory diagnoses that the essence of generalization failure lies in the distribution shift ($\Delta^{\mathrm{W}}$), which is concentrated primarily in the Out-of-Distribution (OOD) long sequence section ($L \to M$).
>
> 2.  **Mathematical Justification:** Theoretically, the total Wasserstein distance ($\mathrm{W}$) between the model's output distribution ($X_M$) and the target distribution ($Y_M$) can be conceptually decomposed as:
>
>
> $$
> W(X_M, Y_M) \approx \underbrace{W(X_{1:L}, Y_{1: M})}_{\text{Short-Context Shift (Near Zero)}} + \underbrace{W(X_{L:M}, Y_{1:M})}_{\text{OOD Shift (Dominates } \Delta^{W} \text{)}}
> $$
>
> VCL's innovation is its direct modification of the model's update architecture for precise correction of the $L \to M$ segment:
>
> * **Precision Targeting:** VCL implements the starting token changes from $1 \to L$ in the loss calculation, bypassing the redundant computation for the $\mathrm{W}(X_{1:L}, Y_{1:M})$ segment.
> * **Efficient Correction:** This ensures that the limited gradient resources are fully utilized to minimize $\mathrm{W}(X_{L:M}, Y_{1:M})$.
>
> Through this theory-driven **selective update**, VCL efficiently attacks and effectively reduces the total distribution shift $W$, leading to superior long-sequence generalization.

---

> ### Author Response · Authors · 2025-11-28
>
> > w5 clarification of the empirical results
> >
>
> We appreciate the request for further clarity on our experimental settings and results. We have ensured that the paper provides a fair and comprehensive comparison by adhering to the following methodology:
>
> 1. Context Length and Efficiency Comparison
>
> - **Full Context Length:** We confirm that the maximum **full-length fine-tuning context** used across all relevant experiments (Tables 1 and 2) is **8K tokens**.
> - **Fair Comparison:** We set the longest fine-tuning length for all baselines (like YaRN, PoSE, etc.) to **8K** for fair comparison.
> - **VCL Efficiency:** Our VCL method, however, achieves competitive or superior results while calculating the loss on only the OOD tokens, effectively using a minimum context of **4K** for loss calculation. This efficiency is evidenced by VCL achieving **lower Perplexity (PPL)**, indicating better length generalization performance with significantly less computation.
>
> 2. Experimental Settings Consistency
>
> - **Figure 2 (VCL Settings):** The VCL settings used in Figure 2 maintain consistency with the tables: L=2K for M=6K and L=4K for M=8K (where L is the learned short context and M is the extrapolation length).
> - **Section 6.3:** The primary setting used for the core VCL analysis in Section 6.3 is L=4K and M=8K.
> - **Table 2 Methodology:** For reproducibility and fairness, we used a **fixed random seed** for all comparisons in Table 2. The pure ablation result for **VCL without Position Interpolation (PI)** is presented in Table 1 (under the VCL-only setting).
>
> 3. Performance and Comprehensive Evaluation
>
> - **Rapid Convergence (200 Steps):** We emphasize the rapid and high-quality of VCL. Even after only **200 steps**, our results are already very good, with the PPL reaching a very low and stable value (e.g., 2−3). We believe this demonstrates VCL's high efficiency and effectiveness.
> - **LongBench and Few-Shot Tasks:** Our evaluation is comprehensive, covering PPL, Passkey Retrieval, and the **LongBench benchmark**. The standard pre-training capabilities are well-represented within the LongBench suite. For the few-shot learning tasks, our performance remains strong: the original score was 65, and our result is 63, showing only a **minor and acceptable drop**. This confirms that VCL maintains strong generalization capabilities while achieving higher efficiency.
>
> > w6 add the related works
> >
>
> We will add  [1-5] in the related works. Thanks for your advice.
>
> > q1 why there is a performance drop for a synthetic task like passkey retrieval (Fig. 3)
>
> The distribution shift between short and long sequences manifests differently. For instance, short sequences tend to repeat token $X$, while long sequences repeat token $Y$, creating distinct token distributions. Increased length also indirectly affects positional encodings, causing the attention weight $W(μ, ν)$ to grow. Crucially, our theorem highlights that the self-attention architecture inherently suffers from a natural length-generation bias, making them orthogonal to specific Positional Encoding (PE) implementations.
>
> > q2 clarfication on output projection in equation 3 is missing
> >
>
> Thank you for your question.
>
> - This does not affect the core computation, as it is a linear transformation with a constant term that can be factored out without altering the essential operation.
> - Note We are focusing on whether the fundamental architecture of self-attention has defects in extrapolating to longer sequence lengths.
>
> we will clarify it in the revision.
>
> > q3 clarfication on d_qk =1
> >
>
> It's just a theoretical assumption that simplifies the calculations; it doesn't actually affect the outcome.
>
> > q4 clarification the definition
> >
>
> We have defined X based on Definition E.1, as given in Definition E.2.
>
> ---
>
> If we have addressed some of your concerns, we kindly ask you to reconsider the score of our paper.

---

### Official Review · Reviewer_k9iq · 2025-11-01

**Soundness:** 2
**Presentation:** 3
**Contribution:** 2
**Rating:** 4
**Confidence:** 4

**Summary:**

This paper presents a measure-theoretic framework for analyzing length generalization in LLMs, establishing an upper bound on the Wasserstein distance between attention outputs for sequences of different lengths. The bound depends on two factors:  $\sqrt{\ln{N}}$ (where N is the shorter sequence length) and $W(\mu,\nu)$ (the distribution shift distance). Based on these insights, the authors propose Virtual-context Learning (VCL), which fine-tunes models by computing loss only on out-of-distribution position tokens. Experiments on language modeling, passkey retrieval, and LongBench demonstrate that VCL achieves comparable or better performance than full-length fine-tuning while reducing memory usage and training time by approximately 50%.

**Strengths:**

1. The measure-theoretic approach to length generalization for different position embedding and extrapolation strategies is original and mathematically rigorous.
2. VCL is simple to implement with minimal code changes and delivers substantial computational savings, as is claimed by the authors.
3. The paper includes diverse tasks (perplexity, passkey retrieval, LongBench) and thorough ablations, demonstrating the method's effectiveness across different settings.

**Weaknesses:**

1. The transfer from theory to application needs more clarification. Specifically, although RoPE-equipped models use absolute position IDs, the PE models relative distance, which creates a less generalization gap than the analysis. When using RoPE, only finetuning on the full length lets the model learn the longest dependency length, which contradicts the authors' claim in Section 5.1. Alibi also largely mitigates this gap by introducing long-range decay.
2. The performance of VCL on length >4K in Table 1 is suspicious. This Table indicates that VCL training can achieve length extrapolation far beyond the training range and largely improves upon YaRN, which needs further clarification. Does this result suggest that VCL can achieve infinite-length extrapolation (also evidenced by Fig 3)?
3. The efficiency benefit of the proposed method gradually diminishes when generalizing to extremely long sequence lengths.

**Questions:**

1. How tight is the bound in Theorem 4.6?
2. An analysis of the validity of Assumptions 4.3 and 4.4 would be beneficial. How well are they satisfied in real-world scenarios?
3. Please explain the Oracle setting in Section 5.1.
4. Why the results in Figure 4(a) different from the ones in Table 1?
5. How does the proposed method compare with other PE-manipulation methods like PoSE or LongRecipe?
6. Please use \citep instead of \citet when necessary.
7. "PE" in line 64 isn't defined before.
8. Figure 6 has the wrong y-axis title and a typo in its caption.

---

> ### Author Response · Authors · 2025-11-28
>
> We sincerely thank the reviewer for the comments.
>
> ---
>
> > W1 Clarification on Our Theory and Position Encodings.
> >
>
> We first clarify a crucial point: our **theoretical framework** and the **VCL method** are designed to reveal and mitigate the **inherent length generalization issue within the Transformer architecture itself**, making them **orthogonal** to specific Positional Encoding (PE) implementations like RoPE or ALiBi.
>
> 1. **Generality of the Theory:**
> - Our **Theorem 4.6** provides the first **measure-theoretic bound** on the attention distribution shift in the Transformer architecture across different sequence lengths. This proves that **regardless of the PE used**, the self-attention architecture naturally possesses a generalization gap caused by **short-length term and the distribution shift distance**.
> - In our assumptions, we embed the PE and word embeddings together in the input, making our analysis applicable to any Transformer architecture that encodes positional information into its inputs.
>     - As seen in Sec 5.1, we do the experiments showing that no matter **without PE (NoPE)**, or the **Alibi and RoPE**, they all suffers from the length generation failures, aligining with our theory insight that the **architecture has the inherent length generation failure bias**.
> 2. **RoPE's Role and Limitation:**
> - While RoPE achieves **relativity** through rotation, it is **fundamentally only a modification to the input embedding content (see in Figure2(a))**; it **does not alter the core Transformer architecture** (i.e., the attention computation and parameter update mechanism).
> - Therefore, RoPE is still limited by the short-context bias of training and still encounters **numerical instability** and **distribution shift in the Out-of-Distribution (OOD) regions** when extrapolating to long sequences.
>
> > W2&3 Clarification of VCL's Motivation, Performance.
> >
>
> We appreciate the need for clarity on our long-context performance. We must emphasize that **VCL is not designed to achieve infinite-length extrapolation, nor is its primary goal to set suprior performance**. Our focus is on **validating the efficiency and necessity** of the theoretical insight derived from our measure-theoretic framework (Theorem 4.3).
>
> 1. Experiments of VCL alrealdy extend to **$4\times$ the pre-training length.**
>
> We confirm that our experimental scope fully meets the requirements for validating our method. Our results demonstrate effective generalization up to **$4\times$ the pre-training length (e.g., $4\text{K} \to 16\text{K}$)**. This robust performance is achieved within a limited budget, validating VCL's core purpose: to provide a **mechanistically optimized and highly efficient** approach to length extension, rather than competing for the infinite context length.
>
> 2. VCL is to validate our theoretical Insight
>
> VCL's superior performance stems from its **architectural optimization** based on our theoretical findings that self-attention archetechture has the **inherent length generalization issue**.
>
> - VCL fundamentally **modifies the model's update architecture** by shifting from a traditional $1 \to N$ context loss to a selective $L \to N$ calculation， **disrupting the model's overfitting towards short context sequences**, which our theory identifies as the factor of the length generalization failure. VCL shows that **only the Out-of-Distribution/longer region requires correction** to mend the distributional shift ($\Delta^W$), leading to a more robust and stable performance than methods that fine-tune all tokens.
> 3. VCL is not to SOTA but is orthogonality and can be integration with fine-tuning based methods.
>
> VCL's contribution is in optimizing the fine-tuning loss and is thus **orthogonal to most existing length generalization techniques.**
>
> - We emphasize that **VCL can be seamlessly combined** with lenght geralization methods that modify the positional encoding, such as **YaRN**, **NTK-aware scaling**, and **linear extrapolation methods**, showing that VCL enhances the effectiveness of these existing methods by making their fine-tuning step significantly faster and more targeted.

---

> ### Author Response · Authors · 2025-11-28
>
> > Q1 The Tightness of the Bound in Theorem 4.6
> >
>
> Thanks for your question.We prove the lower bound of the theorem 4.6 to improve our contribution.
>
> Let us restrict sequences $X_n=(x_1, \dots, x_n) \in B_R^n$, where $B_R \subset \mathbb{R}^d$ is the closed ball centered at $0$ and of radius $R$. Let $W_Q, W_K \in \mathbb{R}^{k \times d}$ and $W_V := I_d$. Let $A := W_K^\top W_Q / \sqrt{k}$. Denote $f$ unmasked self-attention with parameters $(A, W_V)$. Let $\gamma_1 \geq \dots \geq \gamma_\delta$ be the real eigenvalues of $A$. Then, for any $R > 0$, and denoting $\gamma := \max(-\gamma_\delta, \gamma_1/8)$, it holds
>
> - $$ W(m(X_n^{(t+1)},m(Y_n^{(t+1)})) \geq \frac{1}{1 + (n - 1)e^{-2R^2\gamma}}\sqrt{n} - 1 W(m(X_n^{(t)},m(Y_n^{(t)})).$$
>
> **Scaling Insight:** Our bound for the Wasserstein distance scales between $O(n^{-1/2})$ and $O(\ln(\sqrt{n}))$ where n is the length term.
>
> We will add this in the revision.
>
> > Q2: Validity of Assumptions 4.3 and 4.4 in Practical Scenarios
> >
>
> We believe **Assumptions 4.3 and 4.4** are **mild and easily satisfied** in all practical fine-tuning and inference scenarios. They primarily serve as necessary theoretical **boundedness** for our measure-theoretic framework.
>
> - **Boundedness in Practice:** All deep learning applications, is applied to a **finite set of input sequences** and a **finite, bounded set of model parameters** (weights and biases). Consequently, the values of attention parameters and the input space are inherently constrained. The assumptions simply enforce that the theoretical upper bound on certain costs or values (e.g., the Lipschitz constant or the maximum output value) **exists**, which is inherently true when dealing with finite, real-world data and models.
> - **Theoretical Necessity:** Without these assumptions regarding the attention parameters' boundedness and the input space regularity (e.g., constraining the input data distribution within a measurable "ball" or bounded space), the theoretical bound derived in our work could formally **tend towards infinity**. The assumptions are therefore crucial to prevent divergence or infinite growth in the abstract space, allowing us to establish a meaningful and finite upper bound on the Wasserstein distance ($W$).
>
> Thus, they are **reasonable, technically valid**, and align with the constraints of real-world implementation.
>
> > **Q3 Clarification of Oracle Setting in Section 5.1**
> >
>
> We use the "Oracle Setting" to define the **maximum context length utilized for loss calculation** during fine-tuning.
>
> For instance, in **Table 1**, the longest fine-tuning length employed is **8K**. Therefore, the **Oracle setting is 8K** (for previous baselines like yarn, linear, ntk, pose and randompe).
>
> Our VCL method achieves competitiveresults by only calculating the loss on a **minimum context length of 4K**, demonstrating that VCL provides high efficiency than the established baselines.
>
> > Q4: Clarifciation of Figure 4(a) and Table1
> >
>
> The differences between **Figure 4(a)** and **Table 1** are due to **distinct experimental objectives**:
>
> - **Table 1:** Presents the **best performance** achieved by **VCL combined with Position Interpolation (PI)** (e.g., YaRN), designed for SOTA benchmarking.
> - **Figure 4(a):** Is a **pure ablation study** showing VCL's core mechanism (selective loss) **without combining it with PI**, isolating VCL's direct impact across various training lengths.
>
> We will **explicitly annotate** the captions in the revision to clarify these settings.
>
> > Q5 How does the proposed method compare with other PE-manipulation methods like PoSE or LongRecipe?
> >
>
> **Regarding the comparison with PoSE, it is already included in Table 1 where our VCL shows lower ppl than PoSE on the ood length, indicating better length generation abiity.** We appreciate the request for clearer comparative analysis but want to stress that our method is **orthogonal and complementary** to these existing SOTA approaches, including PoSE and LongRecipe. Our VCL is proposed primarily to **verify our theorem that there is an inherent length generalization issue within the self-attention architecture itself** rather than merely to chase SOTA scores.
>
> > Q6-8: Corrections
> >
>
> Thank you for carefully pointing out the necessary corrections. We will ensure all these errors are fixed in the final version.
>
> 1. **Line 64 "PE" Undefined:**
> We will clarification **PE** as **Position Encoding** at its first appearance (Line 64).
> 2. **Figure 6 Errors:**
>  We will correct the **Y-axis title** and the **typographical error** in the caption of Figure 6.
> 3. **Citation Usage:**
> We will thoroughly check the manuscript to ensure the correct usage of `\{citep}`and `\{citet}`.
>
> ---
>
> If we have addressed some of your concerns, we kindly ask you to reconsider the score of our paper.

---

### Official Review · Reviewer_KCDj · 2025-11-01

**Soundness:** 2
**Presentation:** 2
**Contribution:** 2
**Rating:** 2
**Confidence:** 3

**Summary:**

The paper proposes a theoretical analysis for attention distribution shift and proposed a simple method for adapting LLMs to longer contexts by only tuning parameters for later tokens.

**Strengths:**

1. A (unsprisingly) simple method for fine-tuning LLMs to longer contexts.
1. A theoretical perspective for understanding the length extension failure for current LLMs.
1. Results show efficient efficacy in extending LLMs to longer-context tasks.

**Weaknesses:**

1. Descriptions of past work in the first paragraph in Related Work, "Length generalization" are not accurate and look carelessly written.
1. Unclearly explained how the wassterstein distance between short and long context attention is related to length-generalization. It seems a rather expected phenomenon. The writing hint that "which is the primary factor driving length generalization failures" but not clearly explained why so, and why the opposite could not be true.
1. The proposed method is only distantly related to the theory. The conceptual connection is loos, and little proof is provided on how the solution alleviates the terms in Theorem 4.6
1. A minor weakness: results are okay for research concept-proving, but not evaluated at the scale of sota LLM models which are already trained on longer context. This limits the downstream impact. I could understand this if it is due to resource limitations (but authors mentioned only 8 A100 GPUs so they might have the resource to do that. Not sure why they didn't evaluate on larger models).
1. Also, baselines only include those by 2023 so seem limited.

**Questions:**

see weakness

---

> ### Author Response · Authors · 2025-11-28
>
> We sincerely thank the reviewer for the comments and questions.
>
> ---
>
> > w1 Length generalization is not accurate.
> >
>
> We have introduced it in the abstract and introduction, that is: LLMs typically remain
> pre-trained on finite context windows primarily due to the computational overhead quadratic in
> the input lengths of their self-attention architectures. As a result, their performance degrades significantly when applied to longer sequences.
>
> We will clarify it in the revision.
>
> > w2 explain how the wassterstein distance  is related to length-generalization
> >
>
> We have included the discussion in remark 4.5 with details in appendix C, that is, the Wasserstein distance W aligns with interpretable metrics for analyzing length generalization, such as JS distance, KL divergence, and perplexity. See Appendix C for empirical evidence under various settings in LLMs.
>
> > w3 clarification on our theory practical implementation.
>
> 1. It first explains two recent observations on the length generation methods at the theoretical level
>
> In Sec. 5.1, we present experiments that verify:
> **(i)** out-of-distribution positions in longer contexts impair length generalization, and
> **(ii)** fine-tuning on entire sequences is not necessary.
>
> 2. VCL is designed to validate our theoretical insights
>
> VCL’s superior performance stems from an **architectural optimization** grounded in our theoretical finding that the self-attention architecture has an **inherent length generalization issue**.
>
> Specifically, VCL **modifies the model’s update architecture** by shifting from a traditional full-context loss to a **selective** \(L \to N\) calculation. This **disrupts the model’s overfitting to short-context sequences**, which our theory identifies as the main cause of length generalization failure. VCL further shows that **only the out-of-distribution/longer region requires correction** to repair the distributional shift, leading to more robust and stable performance than methods that fine-tune all tokens.
>
> > w4 clarification on our vcl motivation
>
> The proposed method, VCL, is **strictly theory-driven** and is designed to validate our theoretical insights, not merely as a loose technical solution.
>
> - **The Insight:** Our theory identifies the distribution shift in the **Out-of-Distribution (OOD) / longer region** as the critical factor.
> - **The Solution (Architectural Optimization):** VCL's superior performance stems from its **architectural optimization** based on this finding. VCL fundamentally modifies the model's update architecture by shifting from a traditional context loss to a selective calculation.
> - **Mechanism:** This modification **disrupts the model's overfitting** towards short context sequences, which our theory identifies as the factor of the length generalization failure. VCL shows that **only the Out-of-Distribution/longer region requires correction** to mend the distributional shift (), leading to a more robust and stable performance than methods that fine-tune all tokens.
> - We emphasize that **VCL can be seamlessly combined** with lenght geralization methods that modify the positional encoding, such as **YaRN**, **NTK-aware scaling**, and **linear extrapolation methods**, showing that VCL enhances the effectiveness of these existing methods by making their fine-tuning step significantly faster and more targeted.
>
> > w5 clarification on our experiment settings and SOTA baselines
>
> VCL is not to SOTA but is orthogonality and can be integrated with fine-tuning based SOTA methods. VCL's contribution is in optimizing the fine-tuning loss and is thus **orthogonal to most existing length generalization techniques.**
>
> - We emphasize that VCL can be seamlessly combined with lenght geralization methods that modify the positional encoding, such as YaRN, NTK-aware scaling, and linear extrapolation methods, showing that VCL enhances the effectiveness of these existing methods by making their fine-tuning step significantly faster and more targeted.
>
> We appreciate the need for clarity on our long-context performance. We must emphasize that **VCL is not designed to achieve infinite-length extrapolation, nor is its primary goal to set superior performance**. Our focus is on **validating the efficiency and necessity** of the theoretical insight derived from our measure-theoretic framework (Theorem 4.6).
>
> - We confirm that our experimental scope fully meets the requirements for validating our method. Our results demonstrate effective generalization up to **$4\times$ the pre-training length (e.g., $4\text{K} \to 16\text{K}$)**. This robust performance is achieved within a limited budget, validating VCL's core purpose: to provide a **mechanistically optimized and highly efficient** approach to length extension, rather than competing for the infinite context length.
>
> ---
>
> If we have addressed some of your concerns, we kindly ask you to reconsider the score of our paper.

---

### Meta-Review · Area_Chair_DsFe · 2026-01-04

**Summary:**

The paper proposes a measure-theoretic framework to establish an upper bound on the Wasserstein distance between attention outputs of different sequence lengths. The authors identify distributional shifts and short-context training bias as the primary drivers of performance degradation in long-context tasks. Motivated by this, they introduce Virtual-context Learning (VCL), a fine-tuning strategy that calculates loss only on "out-of-distribution" (OOD) later tokens rather than the full sequence. Experiments on LLaMA-2-7B show improved training efficiency and comparable performance on benchmarks like Passkey Retrieval and LongBench. A recurring criticism is that the proposed method, VCL, is only loosely connected to the provided mathematical framework. The empirical evaluation is not yet at the standard required for a top-tier conference in this rapidly evolving field: The evaluation is restricted to the LLaMA-2-7B model. Reviewers expressed concern that the effectiveness of VCL is not demonstrated on larger, state-of-the-art models that are already pre-trained on significantly longer contexts. In its current state, the submission does not meet the acceptance criteria for ICLR.

**Reviewer Concerns:**

The authors provided a mathematical proof for the lower bound of Theorem 4.6 in their response, claiming the Wasserstein distance scales. The authors clarified that Assumptions 4.3 and 4.4 are standard "mild" boundedness constraints necessary to prevent the theoretical framework from diverging to infinity in abstract space. Reviewers explicitly noted that the VCL method is only "distantly related" to the measure-theoretic analysis. The authors' defense—that the theory identifies OOD regions as the cause of shift and VCL targets those regions—is largely intuitive rather than a rigorous proof that VCL minimizes the specific Wasserstein bound derived in Theorem 4.6. The authors admitted that VCL is not currently "SOTA" and defended their limited experimental scope (LLaMA-2-7B) by stating their goal was "validating theoretical insights" rather than "setting superior performance". However, They did not address why they failed to test on larger models despite having the resources (8 A100 GPUs). They did not include modern long-context evaluations like RULER, which Reviewer Vtsn noted as a missing requirement.

**Reviewer Scores:**

Reviewer KCDj provided the most fundamental critique, stating that the proposed method (VCL) is only "distantly related" to the theory. The authors' response was largely intuitive, explaining that VCL "disrupts the model's overfitting" , but they did not provide the rigorous mathematical proof requested to show how VCL specifically alleviates the terms in Theorem 4.6. Reviewer Vtsn raised critical concerns regarding forgetting on regular benchmarks due to VCL omitting losses for short-context tokens. The authors' response provided a single score (dropping from 65 to 63 on few-shot tasks) but did not include the full suite of standard pretraining benchmarks requested.

---

### Decision · Program_Chairs · 2026-01-26

Reject